

# Multi-temporal landslide activity investigation by spaceborne SAR interferometry: Polish Carpathians case study

Kamila Pawluszek-Filipiak[1], Mahdi Motagh[2,3], Andrzej Borkowski[1]

[1]Institute of Geodesy and Geoinformatics, Wroclaw University of Environmental and Life Sciences, Wroclaw, Poland
[2]Institute of Photogrammetry and Geoinformation (IPI), Leibniz Universität Hannover, Hannover, Germany
[3]Hemholtz Center Potsdam, GFZ German Research Center for Geosciences, Potsdam, Germany

*Correspondence to*: kamila.pawluszek-filipiak@upwr.edu.pl

**Abstract.** The main goal of this research is the activity state verification of existing landslide inventory maps using Persistent
Scatterer Interferometry (PSI). The study was conducted in Małopolskie municipality, a rural setting with a sparse urbanization
in Polish Flysch Carpathians. PSI have been applied using Synthetic Aperture Radar (SAR) data from ALOS PALSAR, and
Sentinel 1A/B from different acquisition geometry (ascending and descending orbit) to increase PS coverage and overcome
geometric effects due to layover and shadowing. The Line-Of-Sight PSI measurements were projected to the steepest slope,
which allows to homogenize the results from diverse acquisition modes and to compare displacement velocities with different
slope orientations. Additionally, landslide intensity (motion rate) and expected damages maps were generated and verified
during filed investigations. High correlation between PSI results and in-situ damage observations has been confirmed. Activity
state and landslide-related expected damage map have been confirmed for 43 out of a total of 50 landslides investigated in the
field. The short temporal baseline provided by Sentinel satellite 1A/B data allows increasing of the PS density significantly.
The study substantiates usefulness of SAR based landslide activity monitoring for land use and land development, even in
rural areas.

## 1 Introduction

A landslide inventory map (LIM) is a basic geological and cartographic product providing cumulative information about
landslides. It is the fundamental source for any landslide susceptibility, hazard or risk assessment (Guzzetti et al., 2012). LIM
presents spatial distribution of past and current landslides, their activity state and additional information about their type, size
or even failure mechanisms (Van Den Eeckhaut and Hervás, 2012). LIMs are created using consolidated (field mapping or
visual interpretation of stereoscopic aerial photographs) and innovative (remote sensing) techniques (Guzzetti et al., 2012).
Analysis of panchromatic and multispectral satellite images have been applied to create LIMs using satellite remote sensing
(Martha el al., 2010; Martha et al., 2012; Li et al., 2016).

In the last few decades, differential interferometry SAR (DInSAR) has captured considerable attention in the landslide
community by offering great support for landslide detection and monitoring (Wasowski and Bovenga, 2014). Nowadays,
abundant interferometric synthetic aperture radar (InSAR) application for landslide studies can be found, e.g., landslide





detection (Meisina et al., 2008; Notti et al., 2010; Bianchini et al., 2012; Motagh et al., 2013), landslide characterization (Rosi et al., 2018), landslide monitoring (Notti et al., 2010; Meisina et al., 2008; Haghshenas Haghighi and Motagh, 2016; Herrera et al., 2013) and landslide activity assessment (Righini et al., 2012; Cigna et al., 2013; Bianchini et al., 2013; Frangioni et al.,

2014; Rosi et al., 2018). This is mainly because SAR satellite data provide valuable information about ground displacement for wide area coverage with high spatial resolution and great temporal sampling (Bianchini et al., 2013). To overcome some DInSAR limitations (Wąsowski and Bovenga, 2014), multi-temporal interferometric techniques (MTInSAR) have been developed, e.g., Persistent Scatterer Interferometry (PSI) (Ferretti et al., 2001; Hooper et al., 2004; Kampes, 2006), Small BAseline Subset (SBAS) (Berardino et al., 2002; Crosseto at al., 2008) or SqueeSAR (Ferretti et al., 2011). In the literature,

many diverse MTInSAR techniques exist, which mainly vary with the baseline configuration, the persistent scatterer (PS) selection criteria and the deformation model used among these techniques. A wider overview of these techniques is provided by Sousa et al. (2011) and Crosetto et al. (2016).

Nevertheless, InSAR techniques have some limitations due to the satellite acquisition geometry. Measured deformations using SAR are 1D along the satellite LOS (Line of Sight) direction (Pawluszek-Filipiak and Borkowski, 2019). Therefore, the

fraction of measured deformation depends on the satellite acquisition geometry, topography of the study area and real direction of movements. This is crucial for monitoring landslides (Bianchini et al., 2013). Moreover, because of the ambiguous nature of SAR observations, it is challenging, if not impossible, to measure "fast" deformation (differential deformation > λ/4 over the revisit interval). For example, the maximum measurable deformation for Sentinel 1 and ALOS PALSAR data is 426 and 468 mm/year, respectively (Jia and Liu, 2016). Therefore, InSAR is a suitable technique for monitoring slow and very slow

moving landslides with millimetre precision (Del Ventisette et al., 2014; Mirzaee et al., 2017).

In recent decades, new methods for updating landslide inventory maps by exploiting PSI techniques have been examined by the scientific community. These methods apply traditional thematic maps (geological, topographical and/or optical images) together with field reconnaissance coupled with PSI-delivered ground deformation estimates. PSI techniques are used for (1) landslide phenomena identification (Bianchini et al., 2013; Monserrat et al., 2016), (2) landslide boundaries verification or

modification (Bianchini et al., 2013), (3) landslide velocity and intensity estimation (Bianchini et al., 2012) and (4) activity state assessment (Bianchini et al., 2012; Cascini et al., 2013; Bianchini et al., 2013; Frangioni et al., 2014; Del Ventisette et al., 2014; Kalia, 2018)

Bianchini et al. (2012) presented an application of PSI measurements in a methodology called "Landslide Hot Spot Mapping." The aim of their study was to map and evaluate the state of extremely slow and slow moving landslides. Since then, many

papers have demonstrated the applicability of PSI results for landslide activity assessment. Cascini et al. (2013) applied the PSI approach based on Envisat data, together with damages data for slow moving landslide activity assessment, in southern Italy. Bianchini et al. (2013) applied the PSI approach, based on ALOS PALSAR data, for updating the landslide inventory in Majorca, Spain. Frangioni et al. (2014) applied Envisat data and the PSI-based matrix approach to update the landslide inventory in Setta basin, Italy. The same year, Del Ventisette et al. (2014) applied the PSI matrix approach and ERS 1/2 and

Envisat data to assess landslide activity within the Italian Alps. Commonly used methodology in the abovementioned papers



is the PSI matrix approach with diverse SAR sensors, where specific thresholding of the landslide velocity, acquired from specific PSI processing, are performed. A detailed description of the PSI-based matrix approach is presented in Cigna et al. (2013). Since 2014, when the ESA Copernicus Sentinel 1 satellite was launched, the availability of SAR data has changed for almost entire world. The first articles of using Sentinel-l SAR data for landslide monitoring have been presented by Monserrat

et al. (2016) and Barra et al. (2016). Recently, Kalia (2018) applied 66 Sentinel acquisitions in a descending orbit mode for classification of landslide activity on a regional scale using PSI at the Moselle Valley in Germany.

In reference to the abovementioned papers, the main objectives of the present study are:

•    Updating the pre-existing landslide inventory map in the area of Rożnów Lake in Poland from PSI results derived based on ALOS-PALSAR (2007-2010), Sentinel 1A (2014-2016) and Sentinel 1A/B (2017) data;

•    Evaluating the effect of SAR geometry delivered from ascending and descending orbits from ALOS PALSAR and Sentinel 1 and the sensitivity to measure deformation over the study area;

•    Evaluating the difference in landslide activity updated from three diverse data stacks, namely: L-band (ALOS), C-band with one satellite (Sentinel 1A, with a revisit interval of 12 days) and C-band with two satellites (Sentinel 1A and 1B with a revisit time of 6 days), respectively;

•    Creation of landslide intensity and expected damage maps based on PSI results.

Verification of the objectives and achieved results was done by field investigations. Landslide activity states for 43 out of a total of 50 landslides were investigated. Some examples of landslide activity evaluation inspected during the field work are presented.

This study is probably the first investigation of this type in the Carpathians region. Particularly, the study area is located in the

proximity of Rożnów Lake. This mostly rural area is also attractive for local tourism and rest. Despite the landslide hazard, there is pressure on land development and urbanisation (Kroh, 2017). The realisation of the above-listed objectives can advance the use of the SAR based approach as a reliable information source for future land use and land development.

## 1 Materials and Methods

### 2.1 Study area and existing landslide database

Landslides are common within the area of the Polish Flysch Carpathians. This constitutes a substantial problem for local communities due to the growing population and number of properties located within the landslide prone areas (Crozier, 1986). Around 36% of the Polish Flysch Carpathians are affected by landslides (Gorczyca et al., 2013). Therefore, the area with the most active landslides in Poland has been selected for this research. The region is known for its frequent landslide occurrence. Within the study area, 506 landslides have been identified over past years with at least one landslide per 1 km2. Landslides in

the study area were mapped for the first time in the 1930s (Gorczyca et al., 2013). Since then, the study area has been affected by mass movements several times, mostly triggered by heavy rainfalls. According to historical data, landslides have occurred in the study area seven times between 1934 and 1974 (Gorczyca et al., 2013). More recently, landslides have caused





catastrophic damage due to abundant rainfall and flooding in 1997 and 2010. This region is rural and, thanks to the breathtaking landscapes, the population has grown rapidly in recent decades.

Moreover, it is also attractive to tourists. Apart from scatterer residential areas, many compact tourist buildings can be found in deforested areas. These areas are characterized by high elevations and drainage divides, which are characteristically affected by landslides. In 2010, catastrophic landslide activity damaged approximately 150 buildings and many roads, transmission lines, crops and orchards (Gorczyca et al., 2013). Based on Gorczyca et al. (2013), further landslide activity within the study area is likely, thus, it is important to monitor all landslide activity in this region.

The study area covers the surrounding hills of Rożnów Lake (Fig. 1). This figure shows the landslide distribution within the study area as well their predefined activity states. The first preliminary results of PSI application in the region of Rożnów Lake have been provided by Wojciechowski et al. (2008), Perski et al. (2010) and Perski at al. (2011). Using ERS 1 images, the authors showed the potential of this technique for landslide movement detection in rural and challenging terrain for PSI analysis.

110                                    [Insert Fig. 1 about here]

Figure 1. Study area.

### 2.1.1    Geological and hydrological settings of the study area

From the geological point of view, the study area is located in the Outer Carpathians of the Magura Nappe, Silesian Nappe and Grybów Unit. These geological units are built from flysch rocks, such as clay, sand, gravel, sandstone and shale, of the

Quaternary and Tertiary periods. The study area is located on the borderline between Beskid Wyspowy Mountain and Carpathian Foothills. Beskid Wyspowy comprises low mountains and middle foothills with slopes ranging from 10o to 35o and elevations from 300 to 340 m. The Magura Nappe is highly susceptible to sliding, especially with respect to hieroglyphic-beds containing variegated shales (Burtan et al., 1991; Wójcik et al., 2015). Detailed visualization of diverse geological units is presented in Fig. 2 and the explanation can be found in the supplementary material.

120                                    [Insert Fig. 2 about here]

Figure 2. Geological setting of the study area.

From the geological point of view, the study area is located in the Outer Carpathians of the Magura Nappe, Silesian Nappe and Grybów Unit. These geological units are built from flysch rocks, such as clay, sand, gravel, sandstone and shale, of the Quaternary and Tertiary periods. The study area is located on the borderline between Beskid Wyspowy Mountain and

Carpathian Foothills. Beskid Wyspowy comprises low mountains and middle foothills with slopes ranging from 10o to 35o and elevations from 300 to 340 m. The Magura Nappe is highly susceptible to sliding, especially with respect to hieroglyphic-beds containing variegated shales (Burtan et al., 1991; Wójcik et al., 2015). Detailed visualization of diverse geological units is presented in Fig. 2 and the explanation can be found in the supplementary material.

### 2.1.2 Landslide types and distribution



Among the study area diverse landslide types exist including translational, rotational or combined rock-debris slides and typical debris slides (Pawluszek et al., 2018). A lot of landslides are covered by forest and, therefore, their identification and monitoring presents a significant challenge. The landslide activity in the study area is mostly associated with hydro-geological conditions such as rock stratification and precipitation. These conditions created favorable conditions for landslide activation. Catastrophic landslide activity occurred in 2010 due to cumulative rainfalls ranging from 50 to 400 mm over 2 to 5 days

activating mudslides and debris slides. Activation of deep rockslides required long continuous precipitation of 100 to 500 mm per month. Most of the landslide scarps within the study area lead down to valley floors or river terraces where the colluvium consists of weathering debris, clay with debris and pure clay. Usually, north-facing landslides tend to be complex, while south-facing landslides tend to be insequent or subsequent slides (classification based on the relationship to the arrangement rock layers as described in (Gorczyca et al., 2013).

**2.1.2 Landslide types and distribution**

Among the study area diverse landslide types exist including translational, rotational or combined rock-debris slides and typical debris slides (Pawluszek et al., 2018). A lot of landslides are covered by forest and, therefore, their identification and monitoring presents a significant challenge. The landslide activity in the study area is mostly associated with hydro-geological conditions such as rock stratification and precipitation. These conditions created favorable conditions for landslide activation.

Catastrophic landslide activity occurred in 2010 due to cumulative rainfalls ranging from 50 to 400 mm over 2 to 5 days activating mudslides and debris slides. Activation of deep rockslides required long continuous precipitation of 100 to 500 mm per month. Most of the landslide scarps within the study area lead down to valley floors or river terraces where the colluvium consists of weathering debris, clay with debris and pure clay. Usually, north-facing landslides tend to be complex, while south-facing landslides tend to be insequent or subsequent slides (classification based on the relationship to the arrangement rock

layers as described in (Gorczyca et al., 2013).

**2.1.2 Pre-existing landslide inventory map**

The Landslide Counteracting System (SOPO) project was used for the pre-existing landslide inventory database. LIMs are the components of this database. The SOPO project was launched in 2008 under the order of the Ministry of Environment with funds from the National Fund for Environmental Protection and Water Management. The purpose of SOPO is to support the

administration and environmental protection inspectorates, as well as non-governmental organizations, in effectively fulfilling the duties connected to landslide risk management. The system, in its assumption, provides proper and complete data for effective landslide risk management (Pawluszek et al., 2018). Within the SOPO project, all recorded landslide locations and their extent have been stored. Moreover, information about activity state as well as detailed geomorphological and geological analysis, are presented. Landslides within the study area were mapped during field work in 2010, 2011, 2012, 2013, 2014 and

2015 (Bąk et al., 2011; Wójcik, et al., 2011; Koluch and Nowicka, 2012). Additional mapping work was also performed on



the basis of topographic maps at 1:10,000 scale and supported by stereoscopic analysis of aerial photographs and LiDAR data (Perski et al., 2011; Wojciechowski et al., 2012).

## 2. Methods

The methodology flowchart is presented in Fig. 3. The generation of landslide activity maps integrates the PSI data, the PSI
post-processed delivered products and landslide inventory map. These three consecutive stages are described in the following subsections.

[Insert Fig. 3 about here]

Figure 3. Methodology.

### 2.1 Radar data and PSI processing

Table 1 presents five different data stacks, which have been processed using PSI. Due to the limitation of traditional DInSAR techniques related to degradation of radar signals resulting from atmospheric effects and temporal and geometrical decorrelation, the PSI approach has found widespread application in landslide studies (Perski et al., 2011; Bianchini et al., 2012). The PSI approach is based on exploration of stable natural or man-made reflectors which present stable backscattering signal over time. Actually, the PSI approach has been previously applied in monitoring Carpathian landslides (Perski et al.,
2010, 2011). However, due to the high temporal decorrelation, resulting from vegetative cover and short wavelength (X-band), it was only partially successful (Perski et al., 2009; Perski et al., 2011). This is mostly related to low PS density due to the TerraSAR-X data application. Therefore, exploitation of C-band (Sentinel-1) and L- band (ALOS PALSAR) data can bring more advantages, especially in rural areas (Lu et al., 2018).

[Insert Tab. 1 about here]

180                                        Table 1. Data stacks used and their metadata.

In this work, we exploited the PSI approach introduced by Feretti et al. (2001). This approach utilized more than 20 images to separate diverse interferometric components which correspond to deformation components, atmospheric error or topographic error due to the difference between digital terrain modelling (DTM) used for topography estimation and real surface backscattering elevation. The PSI approach is based on the following steps: (1) interferogram generation with respect to a
common master image, (2) PS candidate selection based on amplitude dispersion index, (3) first estimation of atmospheric phase screen (atmospheric influence) and topographical and displacement components and (4) second estimation of interferometric components and final PS points selection. For precise description of the algorithms applied within each step, see Ferretti et al. (2000, 2001).

The present approach was performed for ALOS and Sentinel data stacks independently. Some necessary pre-processing steps
were applied for raw SAR data, namely, focusing, image cropping and compensating for zero Doppler centroid. SARscape®



software was used for SAR data processing (Sahraoui et al., 2006). Numerous interferograms were created from single look complex (SLC) data in original resolution without applying any pixel averaging. The precise orbital files from the European Space Agency (ESA) were used and SRTM 3 sec digital elevation models3 sec with resolution of 90 m were applied for topographic phase removal.

**2.2 PS post-processing phase**

After PSI processing, all results for the five diverse data sets have been post-processed in order to retrieve the most adequate displacement information (see module 2 in Fig. 3). Subsequent post-analysis descriptions are provided in the following subsections.

**2.2.1 PS suitability analysis**

PSI measurements provide deformation information in the LOS direction (from the satellite to the ground as determined by the incident and heading angles). Thus, it is impossible to retrieve 3D displacement from InSAR directly. Horizontal and vertical components of movement (assuming no N-S horizontal motion exists within the study area) can be retrieved by combining measurements from ascending and descending orbits. However, in mountainous areas, where PS coverage is low due to the geometrical conditions and distortions, velocity decomposition can be problematic. Therefore, conversion of LOS

deformation into the most probable direction (direction of maximum slope), by assuming a pure translational movement mechanism, is commonly used (Bianchini et al., 2012). In order to assess the sensitivity of InSAR LOS measurements in a specific point, we applied R index. This index represents the geometrical visibility of the specific area with respect to the SAR system used. R index takes into account morphology of the study area and the acquisition geometry of the particular SAR system and is calculated as ( Notti et al. (2010) ):

$$R = -\sin(S \cdot \sin(A - \alpha + 90) - \vartheta) \tag{1}$$

where $\alpha$ is the angle of the satellite track from North, $\vartheta$ is the incidence angle, S is the slope and A stands for the aspect. The calculated R index can take values within the range [-1, +1].

Figure 4 provides the visualization of the R values over the study area for ALOS and Sentinel imaging systems.

215                                    [Insert Fig. 4 about here]

Figure 4. R index for three diverse satellite geometries (ALOS ascending and Sentinel ascending and descending modes).

R index close to zero shows low SAR sensitivity to measured displacements for specific locations. Based on testes for different test sites, Notti et al. (2014) have found out that distinguishing of four classes is the optimal solution.  Therefore, R index has

been divided into four classes, wherein we used the same interval length for all classes except the first. The first class (R index <0) represents the location where detection of the PS points is difficult due to geometrical distortions. The Second class (R index 0 to +0.33) indicates the location where the slope geometry is not convenient for SAR measurements. The third group



(R index 0.33 to 0.66) presents PS on slopes with acceptable geometry for SAR displacement measurements. The fourth group (R index 0.66 up to 1) represents high sensitivity of LOS deformation monitoring. This means that, in this location, the slope direction is almost perpendicular to the satellite LOS direction and SAR sensors can perfectly measure the deformation over this location. Summarizing, in further studies, the PS points with R index higher than 0.33 were considered.

**2.2.2 PS velocity projection along the steepest slope**

Deformation retrieved from PSI is 1D measurement in the direction to the satellite. LOS deformations ($V_{LOS}$) were projected along the steepest slope according to the equation:

$$V_{slope} = V_{LOS}/\cos\beta \qquad\qquad (2)$$

with β as the angle between the steepest slope and the LOS direction.

Despite the great advantage of the motion represented in the slope direction, this projection has some limitations. First, when β=90o, Vslope goes into infinity. Here we followed Herrera et al. (2013) and selected an absolute maximum value of β=72o, which is equivalent to cosβ=0.3, so that Vslope cannot be higher than 3.33 times than that of VLOS. In order to remove any Vslope exaggeration, we considered only PS points for which cosβ >0.3.

Moreover, we discarded PS points which show Vslope>0 because positives values represent uphill movements and it is not representative for small landslide movements, even though positive values exist within landslides, especially in the toe area.

**2.2.3 Velocity thresholding for activity state estimation - PSI based matrix approach**

For the activity and intensity assessment, some representative values have been retrieved from PSI analysis. These values are fixed values which are used for: (1) distinction between moving and non-moving landslides and (2) discrimination of extremely slow from very slow moving landslides. It strictly depends on the study site considering the deformation processes and typology (Cigna et al., 2013). Commonly, the average of LOS velocity estimates is applied as representative of velocity (Bianchini et al., 2012). However, different thresholds are applied to assess the landslide activity. These values can vary with respect to SAR data wavelength and projection of the velocity. For VLOS, some authors (Righini et al., 2012; Herrera et al., 2013) utilized 2 mm/yr as the velocity threshold for landslide activity assessment for C-band data and 5 mm/yr for Vslope (Cigna et al., 2013, Bianchini et al., 2013). These changes are mostly correlated with different velocity distribution patterns. For the LOS velocity, distribution is almost normal (Gaussian), while for SLOPE is second negatively skewed as a result of the PS reduction (Bianchini et al., 2013). Therefore, for activity state estimation, we applied 5 mm/yr as the Vslope threshold.

Based on the pre-existing landslide inventory map and PSI post-processed results, we applied the PSI-based matrix approach (Fig. 5). According to the PSI matrix (Fig. 5), the presented approach employs a simplified version of the official categorization of the landslide activity state presented in the landslide glossary WP/WLI, (1993). Four diverse activity states have been determined (Fig. 5): (1) reactivated = active after being inactive, (2) active continuous = currently moving, (3) dormant = inactive, but possible to be reactivated and (4) stabilised = not active anymore.




[Insert Fig. 5 about here]

255                Figure 5. PSI based matrix activity assessment (based on Cigna et al., 2013).

### 2.2.4     Landslide intensity estimation

Based on the representative values of deformation, landslide intensity was assessed relying on the Cruden and Varnes (1996) intensity scale. This scale is based on PSI velocities consisting of three categories: negligible, extremely slow and very slow (Fig. 6). In reference to Cruden and Varnes (1996), landslides with sufficient information are divided into negligible (mean

velocity 5 mm/yr), extremely slow (mean velocity between 5 and 16 mm/yr) and very slow (mean velocity between 16 mm/year and 1.6 m/yr). It is also worth to mention, that some researchers use slightly lower thresholding (e.g. Bianchini et al., 2012) with the argumentation that PSI underestimates movements in comparison to the real movements.

[Insert Fig. 6 about here]

Figure 6. Classification of PSI delivered movement rates for landslide intensity scale estimation. $V_p$ and $V_h$ are present or historical PSI estimated movements, respectively.

### 3.     Results

### 3.1     Landslide activity state and intensity map generation

The landslide activity and intensity maps integrate the post-processing derived products ($R$ and $V_{slope}$) with the existing

landslide inventory. For the assessment of landslide activity, the previously described PSI matrix-based approach was applied. Results of the activity assessment are presented in Fig. 7. However, the activity state has been presented only for landslides where sufficient PS points have been found. At least four PS points within a landslide body were set up as the threshold. A number of landslides in the study area are complex landslides. Such landslides are divided into parts and represented in SOPO databases as separate objects. In this case, the mentioned threshold is related to each landslide object.


[Insert Fig. 7 about here]

Figure 7. PS points after post processing phase (on the left) and landslide activity state assessment (on the right) for three diverse data stacks processing results.

In Fig. 8, the landslide intensity scale was estimated based on the abovementioned classification (subsection 3.4).

[Insert Fig. 8 about here]

Figure 8. Landslide intensity scale estimation based on ALOS, Sentinel 1A and Sentinel 1A/B delivered PSI results.



After the post-processing phase, 3898, 5260 and 10,798 PS points were found within the landslide boundaries for ALOS
PALSAR, Sentinel 1A and Sentinel 1A/B data stacks, respectively. This allowed us to estimate the activity state and intensity
scale for 128, 130 and 205 landslides using ALOS, Sentinel 1A and Sentinel 1A/B data, respectively.

Fig. 9 presents, in the form of pie chart some statistics to landslide activity state assessment depicted in Fig. 7. The left column
of pie charts in Fig. 9 corresponds to the maps shown in Fig. 7 (a), (c), (e) respectively. From the total number of landslides
and landslides objects (according to SOPO database), 23%, 20% and 38% have been updated by PSI approach for ALOS,
Sentinel-1A and Sentinel-1A/B data respectively. Percentages of particular activity states within updated landslides are
illustrated in the right column of pie charts in Fig. 9. These pie charts correspond to the maps in Fig 7 (b), (d), (f). Note, that
the last maps show also activity state updated based on PSI-matrix. Therefore there are also landslides, which activity state
has been updated based on historical or pre-existing data (SOPO database) if insufficient number of PS were detected on the
landslide object (compare also Fig. 5).

[Insert Fig. 9 about here]

Figure 9. Percentage of updated landslides

### 3.2. Possible hazard assessment

Based on a literature review, a downstream investigation was performed and additional thresholds were set up in order to
assess possible hazards related to buildings and infrastructure located in landslide areas. For this purpose, we applied the
method proposed by Mansour et al. (2011), i.e., the threshold of 10 to 100 mm/yr as a minimum landslide velocity which can
cause moderate damage to infrastructure and buildings. Velocity rates higher than 100 mm/yr can cause major damage to
infrastructure and buildings. Landslide with velocity below 10mm we classified as landslide with minor expected damages.
This thresholding has been adopted as an additional criterion in order to support environmental planning and management
strategies to areas which can be characterised by high landslide hazard and, consequently, should be addressed to potential
damages protection. In Fig. 10, possible damages caused by mass movements in the study area are presented for three diverse
PSI processing results.

[Insert Fig. 10 about here]

Figure 10. Expected damages from landslides based on Mansour et al. (2011) and PSI results from ALOS, Sentinel 1A and
Sentinel 1A/B data



## 3.3 Field validation

The confidence degree of landslide activity maps can be validated throughout their comparison with external independent
sources, such as damage inventories, *in situ* measurements, field checks, etc. This evaluation aims at assessing whether
measured displacements represent landslide dynamics. It is worthwhile to emphasise that the reality of the PSI-velocity
estimation, itself, is not assessed due to the lack of external field measurements, but whether this measurement is related to
landslide activity or not. For this reason field verification was performed.

Due to the abundant number of landslide located within the study area (506), only 50 landslides, which are expected to produce
moderate damage, were investigated in the field. Activity states have been confirmed for 43 landslides, which indicates around
86% success rate. However, even if no evidence was found in the field, it does not mean that the landslide is not active.
Landslide damage, especially in agricultural areas and to buildings, are immediately repaired and, therefore, not always visible
to people living there. Some examples of field verification and descriptions of investigated landslides are presented in the
following subsection. The field work was performed on November, 2018 and was led by an experienced landslide expert.

**Landslide "Just-Tęgoborze," SOPO ID 23374**

The landslide "Just-Tęgoborze" is a group of rock and debris slides occurring below St. Just pass. This landslide is located on
the southern and eastern slopes. The landslide was developed on clay of various genesis and from the marginal Magura Nappe
formations. The upper part is founded on the outcrops of the Magura sandstones and the lower part on the slate sandstones of
the sub-Magura units. It has been documented as active for the last 50 years. The activity state is regularly confirmed by road
damage. Essential movements occurred in June 2010 as a result of abundant rainfall. In addition, many residential buildings
have been damaged. Landslides tend to develop and increase activity over a large area. Presented landslides are very difficult
to stabilise due to the activity, considerable thickness of the colluvium and slate shales occurrence in the ground (Wójcik et
al., 2011). This landslide was assessed as active (in 2010), dormant (2014-2016) and reactivated (in 2017) based on the PSI-
base matrix approach. Within the landslide area, there were moderate damages in 2010 and 2017 and minor damages in 2016
based on Mansour et al. (2011). In Fig. 11, the extent of the Just-Tęgoborze landslide and PS points are presented together
with photographs taken during the field verification.

[Insert Fig. 11 about here]

Figure 11. Just-Tęgoborze landslide activity evaluation during field trip.

**Landslide "Zbyszyce" SOPO ID 73253**

The Zbyszyce landslide is located in the Zbyszyce village in the Gródek nad Dunajcem commune on the northwestern slope
of the Dąbrowska hill (581 m above sea level). It covers the whole slope up to the bottom of the Dunajec River Valley, where
the river forms a delta. The landslide is located on the overlapping area of Magura and Dukla Nappes. The Magura units





occupy the largest area under colluvial sediments and are represented by thin and medium-sized dominant sandstones and clay

and mud shale layers of larchwort belonging to stratiflycam (Santon-Paleocene). Above them, in the dorsal parts, there are speckled slates overfilled with thin sandstones (Paleocene-Eocene). In the south and southeastern directions, these settlements are also flat on the pieces, which belong to the Dukla unit. In the southern part of the landslide, there are thick-walled Cergowa sandstones (from Oligocene) and in the eastern part sandstones and shales of the Lower Krosno layers (Perski and Wójcik, 2010a; Wojciechowski et al., 2012).

In addition to the discontinuities associated with the slides, the geological structure of the landslide is marked by faults. The Zbyszyce landslide has been active for at least 60 years. The communal road crossing the landslide is constantly destroyed in the middle and lower parts. This landslide was assessed as active (in 2010), dormant (2014-2016) and reactivated (in 2017) based on the PSI-based matrix approach. Within the landslide area, there were expected moderate damages (in 2010 and 2017) and minor damages (in 2016). In Fig. 12, the extent of the Zbyszyce landslide and PS points are presented together with

photographs taken during the field verification and documentation of active landslide conditions.

[Insert Fig. 12 about here]

Figure 12. Zbyszyce landslide activity evaluation during field trip.

**Landslide "Lipie-Jelna" SOPO ID 73194**

The Landslide "Lipie-Jelna" landslide is located between the towns of Lipie and Jelna. The landslide is a rock and debris slide. The landslide material consists of clays, sandstones and shales from the Quaternary period. It is an old landslide with main scarps located more or less parallel to the extent of the landslides, ending with a 3 m landslide crown. The landslide area covers the whole area of the slope. When reactivated, this landslide covered about 80% of the landslide area. It tends to develop uphill and its activity depends on weather conditions. Many building cracks, as well as damage to roads and cultivated fields exist

within the area of the landslide (Wójcik and Krawczyk, 2010). This landslide has been assessed as active based on all PSI results with moderate damage caused by landslide activity. In Fig. 13, the extent of the Lipie-Jelna landslide is presented together with photographs taken during the field verification.

[Insert Fig. 13 about here]

Figure 13. Lipie-Jelna landslide activity evaluation during field trip.




**Landslide "Wola Kurowska" SOPO ID 73254**

The Wola Kurowska landslide is located on the eastern slope of the ridge, 418.8 m above sea level and on the eastern part of the provincial road. There is a rock and debris slide that probably was created in late Holocene. The landslide is built of clay,

sandstone and shale. The main scarp of the landslide is heavily obliterated. The landslide is developed in the Magura Nappe and the Silesian Nappe. As a result of continual rainfall in 2010, 20 m of provincial road and two buildings with agricultural areas were damaged. This landslide was probably active earlier as is indicated by the ruins of older buildings (Perski and Wójcik, 2010b). This landslide was assessed as active (in 2010), dormant (2014-2016) and reactivated (in 2017) based on the PSI-based matrix approach. There are expected moderate damages in 2010 and 2017 and minor damages in 2016 due to

landslide activity. In Fig. 14, the extent of the Wola Kurowska landslide is shown together with photographs taken during the field verification. It is clearly visible correlation between the assessed landslide activity and damages.

[Insert Fig. 14 about here]

Figure 14. Wola Kurowska landslide activity evaluation during field trip.

**Landslide "Bartkowa-Posadowa" SOPO ID 72917**

The Bartkowa-Posadowa landslide is a rock and debris slide covering almost all parts of the slope. Characteristic landslide features are destroyed and transformed by intensive agricultural activity. This is a frontal landslide which means the width is greater than the length. The main slope has the tendency to develop up the slope which causes damage to the commune road. On this road, there are transverse faults and a drop of up to 0.5 m, and the depressions are repaired by leveling the surface.

Other communal roads have also been destroyed.

During the field check, numerous forms (scarps, cracks) characteristic of active landslides were found. The landslide is active over almost the whole area. There are more than 20 residential buildings with visible damage to walls and floors. It is not possible to stabilise the entire landslide, as well as fragments of landslides and current landslide processes, due to the depth of the slip surface (up to 22 m) (Chowaniec et al., 2012). Destroyed buildings are not suitable for renovation, and residents should

be resettled. This landslide was assessed as active (in 2010), dormant (2014-2016) and reactivated (in 2017) based on the PSI-based matrix approach. There were moderate damages in 2010 and 2017 and minor damages in 2016 based on Mansour et al. (2011). In Fig. 15, the extent of Bartkowa-Posadowa landslide is visualised together with photographs taken during the field verification.

400                                        [Insert Fig. 15 about here]

Figure 15. Bartkowa-Posadowa landslide activity evaluation during field trip.





# 4        Discussion

The outcomes of conducted investigations suggest that the PSI technique has great potential in the field of landslide activity
and intensity assessment. Moreover, PSI results can be applied in security management for assessing potential hazards related
to landslide occurrence. However, PSI application needs to be carefully evaluated. Even if the use of the PSI technique can
overcome some of the limitations of conventional monitoring techniques (e.g., large coverage of the area to be monitored, high
costs, terrain inaccessibility, installation problems and maintenance), in many cases, assessment of landslide activity might
become very challenging. This is mostly connected with PS density in rural areas, where temporal decorrelation affects PS
density. However, it was demonstrated that increasing the temporal sampling rate in view of launching the second Sentinel 1
satellite provides higher PS density and, ipso facto, more landslides could be investigated by means of PSI. We were able to
investigate the landslide activity states of 130 and 205 landslides by using one Sentinel satellite and two satellites, respectively.
It is significant progress in comparison to the first attempts with the application of PSI technique using TerraSAR-X and ERS-
1 data in this study area (Perski et al., 2010, 2011) that failed due to disturbing vegetation cover.

From another point of view, application of two sentinel datasets from ascending and descending orbits using only one Sentinel
satellite (C-band) allowed us to achieve similar PS density in comparison to one ascending dataset of ALOS PALSAR (L-
band).For both sensors, ALOS and Sentinel 1 in ascending geometry, the slope movement in a NE-SW direction represents
only a small percentage of the real occurred displacements in LOS displacement rates measured with the interferometric
analysis. This means that many landslides which are facing mostly north-east or north-west slopes cannot be monitoring using
interferometry with ascending acquisition geometry. For these slopes, descending geometry is more suitable, thus deformation
monitoring from both acquisition modes should be combined. Therefore, the projection of the VLOS along the steepest slope
permits us to homogenise landslide velocity achieved from PSI processing from diverse orbit modes. Moreover, many of the
observed landslides have sensitivity lower than one, which means that the real displacement rate can be much higher than those
observed by interferometric techniques.

In summary, the PSI analysis for both sensors, namely ALOS PALSAR and Sentinel images, reveals quite satisfactory
performance for assessing the landslide activity in the rural areas in the Polish Flysch Carpathians. ALOS PALSAR, thanks to
the L-band sensor with high penetration capacity, and Sentinel 1, thanks to its high temporal sampling (6 days), offers a great
advantage in monitoring ground displacements from satellites. However, based on the abovementioned limitation connected
with underestimation of displacement rate, it is recommended that achieved results be evaluated with in situ methods or field
validation. These methods should be complementary to each other. The results retrieved by SAR interferometry can be
considered as the initiation of the field inventory. Therefore, in in this study, landslide activity states were evaluated by field
reconnaissance. All landslides presented as examples of field verification, based on Mansour's thresholding, are expected to
generate moderate damage to buildings and infrastructure. This fact was confirmed in the field, where the evidence of landslide
activity was found by observing ground and infrastructure damage. In this context, the results of our investigations can leverage





the application of the PSI approach in land use and land development studies. It can also support mentioned by Kroch (2107) reduction of the urbanization pressure in the Roznowskie Lake region.

### 5. Conclusions

In this research, landslide activities were studied based on PSI analysis conducted on datasets delivered from three SAR-satellites, namely, ALOS PALSAR, Sentinel 1A and Sentinel 1A/B. The landslide monitoring was carried out in the area of the Polish Flysch Carpathians in Małopolskie municipality, which has been widely affected by landslides for decades.

The landslide activity states and intensities were assessed based on the PSI-based matrix approach. The PSI analysis consisted of five diverse PSI processing for ALOS ascending mode and Sentinel ascending and descending modes (2014-2016), and for Sentinel ascending and descending modes (2017). Deformation in LOS direction were projected onto the steepest slopes. This allowed us to combine ascending and descending results together as one PSI result. Additionally, for each PSI result, potential damage maps were created based on Mansour at al. (2013) criteria. Landslide activity states were also evaluated in situ. Some examples of field evaluation have been presented together with general landslide characteristics.

In general, applying the PSI approach in rural areas is challenging. However, the results of this study prove that increasing the temporal sampling rate in view of launching the second Sentinel 1 satellite, provides higher PS density and, therefore, this technique can deliver useful information for landslide activity assessment. Sentinel 1A and Sentinel 1A/B were used for 130 and 205 landslides, respectively.

From another point of view, application of two Sentinel datasets from ascending and descending orbits, using only one Sentinel satellite (C band), allows us to achieve similar PS density in comparison to one ascending dataset of ALOS PALSAR (L-band). Overall, the outcomes of this work justify the potential of the PSI approach for landslide activity and intensity assessment. We have shown that this technique can be beneficial for downstream applications related to landslides.

### Acknowledgments

This work was carried out by Kamila Pawluszek-Filipiak during a research internship in the German Research Center of Geosciences in Potsdam within the project "Innovative Doctorate" (No. D220/0001/17) founded by Wroclaw University of Environmental and Life Sciences. The authors are very grateful to Tomasz Wojciechowski, Zbigniew Perski and Piotr Niescieruk for providing geological information, photographs and documentation of the study area and for their support during field investigations accompanied by enlightening lectures. Furthermore, the authors are also grateful to Hans-Urlich Wetzel for his valuable discussions connected with landslide geological issues.

### Appendix A. Supplementary Material

Explanation of Geological Units





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

**Table 1. Data stacks used and their metadata.**


|  | *Data stack 1* | *Data stack 2a* | *Data stack 2b* | *Data stack 3a* | *Data stack 3b* |
|---|---|---|---|---|---|
| *Satellite* | ALOS PALSAR | Sentinel 1A | Sentinel 1A | Sentinel 1A+1B | Sentinel 1A+1B |
| *Wavelength* | L-band | C-band | C-band | C-band | C-band |
| *Repeat cycle [days]* | 46 | 12 | 12 | 6 | 6 |
| *No. of images* | 22 | 56 | 52 | 60 | 53 |
| *Orbit mode* | ascending | ascending | descending | ascending | descending |
| *Spanning interval* | 31/01/2008-27/12/2010 | 29/11/2014-24/12/2016 | 8/12/2014-21/12/2016 | 5/01/2017-31/12/2017 | 2/01/2017-22/12/2017 |
| *Polarization* | VV | VV | VV | VV | VV |



| Incidence angle | 38.7 | 39.05 | 33.71 | 33.04 | 33.71 |
|---|---|---|---|---|---|
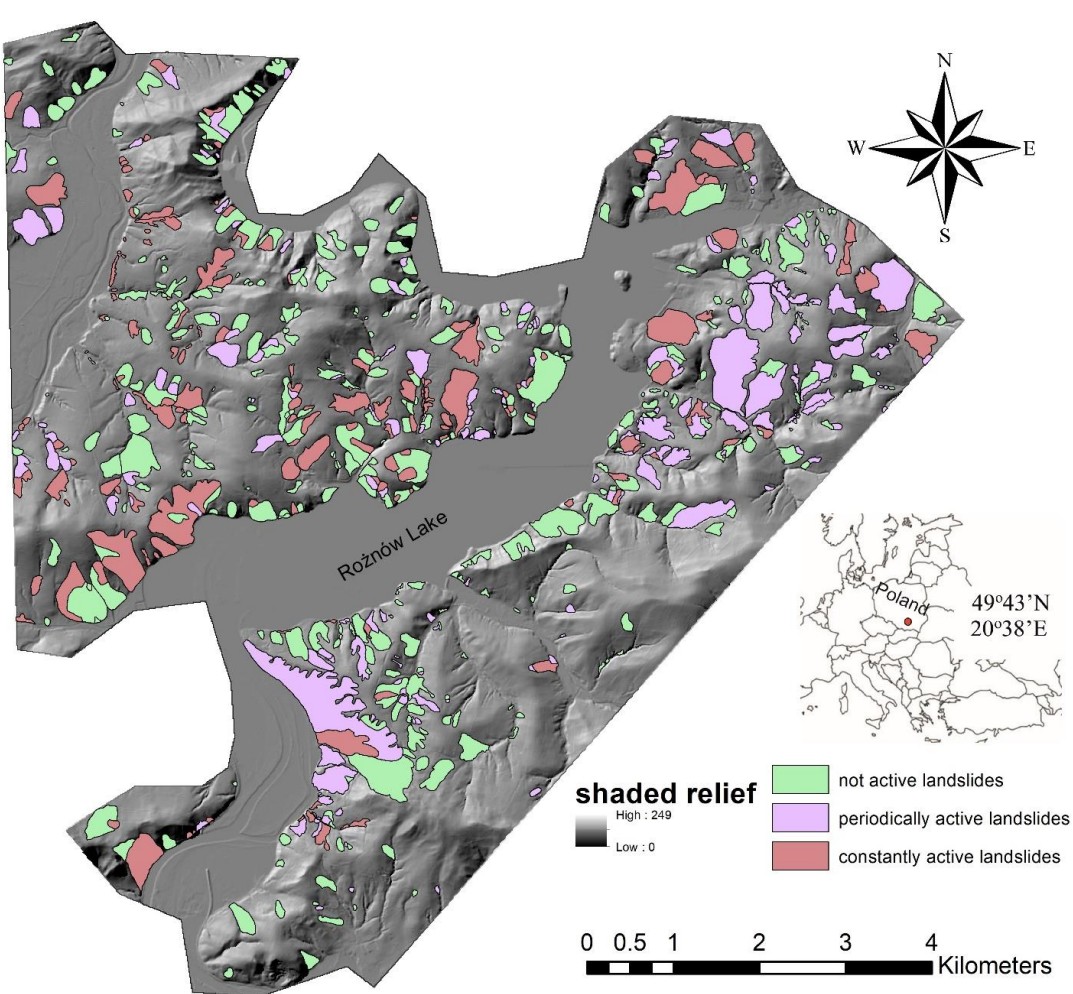

**Figure 1: Study area.**






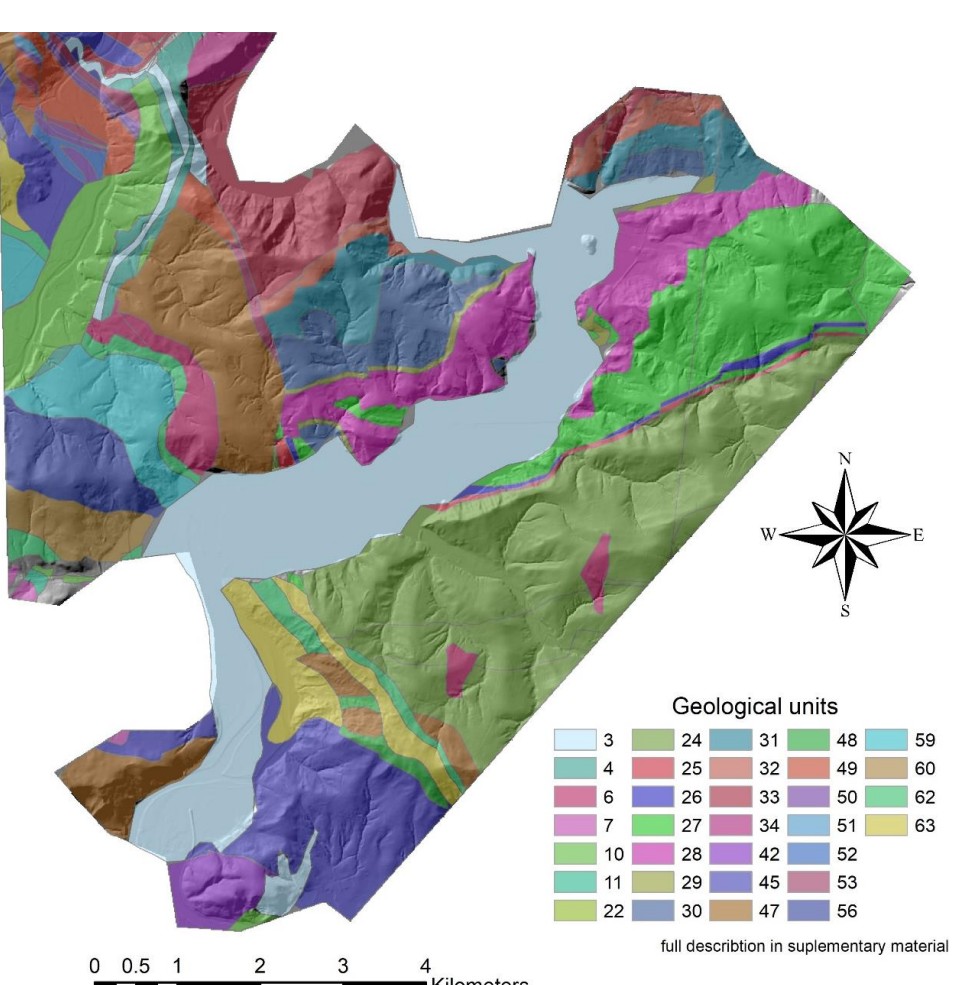

**Figure 2: Geological setting of the study area.**




**Figure 3. Methodology.**





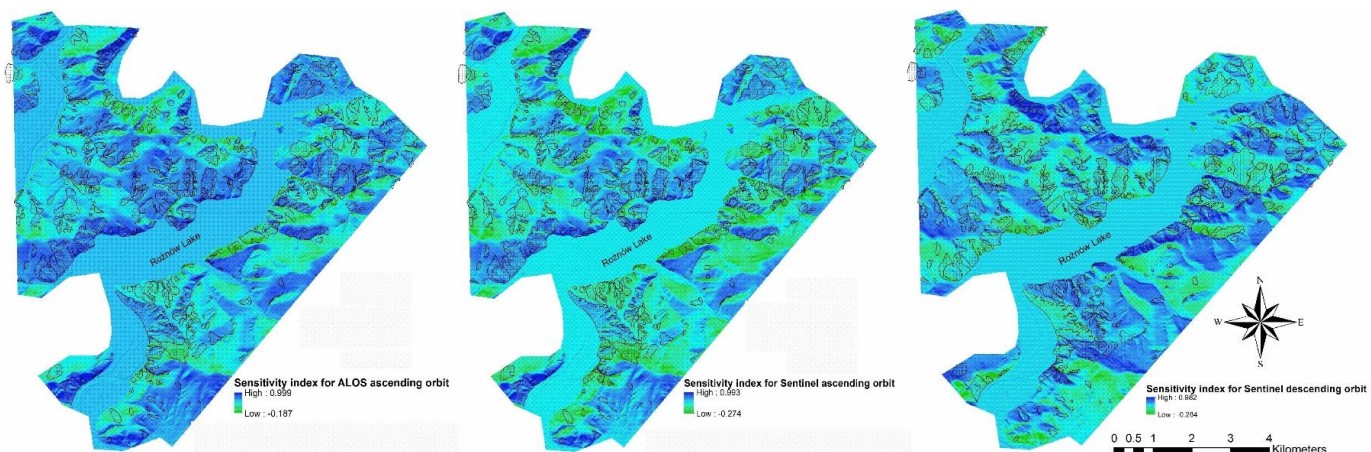

**Figure 4. R index for three diverse satellite geometries (ALOS ascending and Sentinel ascending and descending modes).**




**Figure 5. PSI based matrix activity assessment (based on Cigna et al., 2013).**

**Figure 6. Classification of PSI delivered movement rates for landslide intensity scale estimation. Vp and Vh are present or historical PSI estimated movements, respectively.**

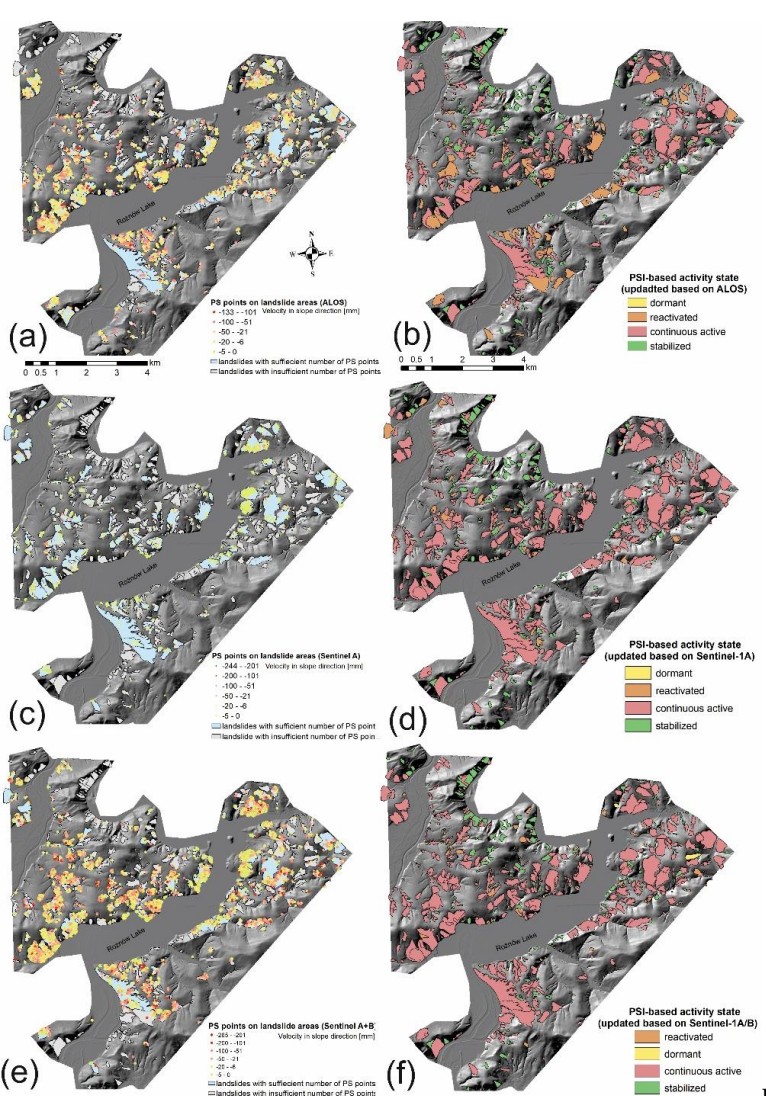

**Figure 7. PS points after post processing phase (on the left) and landslide activity state assessment (on the right) for three diverse data stacks processing results..**



Figure 8. Landslide intensity scale estimation based on ALO (a) S, Sentinel 1A (b) and Sentinel 1A/B (c) delivered PSI results

**Landslide intensity
(based on Sentinel-1A)**

negliglible

extremely slow

very slow

insufficient number of PS

(b)

Rożnów Lake

**Figure 8. Landslide intensity scale
estimation based on ALO (a) S, Sentinel 1A (b) and Sentinel 1A/B (c) delivered PSI results.**



**Landslide intensity
(based on Sentinel-1A/B)**

☐ negliglible
☐ extremely slow
☐ very slow
☐ insufficient number of PS

(c)

**Figure 8. Landslide intensity scale**
estimation based on ALO (a) S, Sentinel 1A (b) and Sentinel 1A/B (c) delivered PSI results



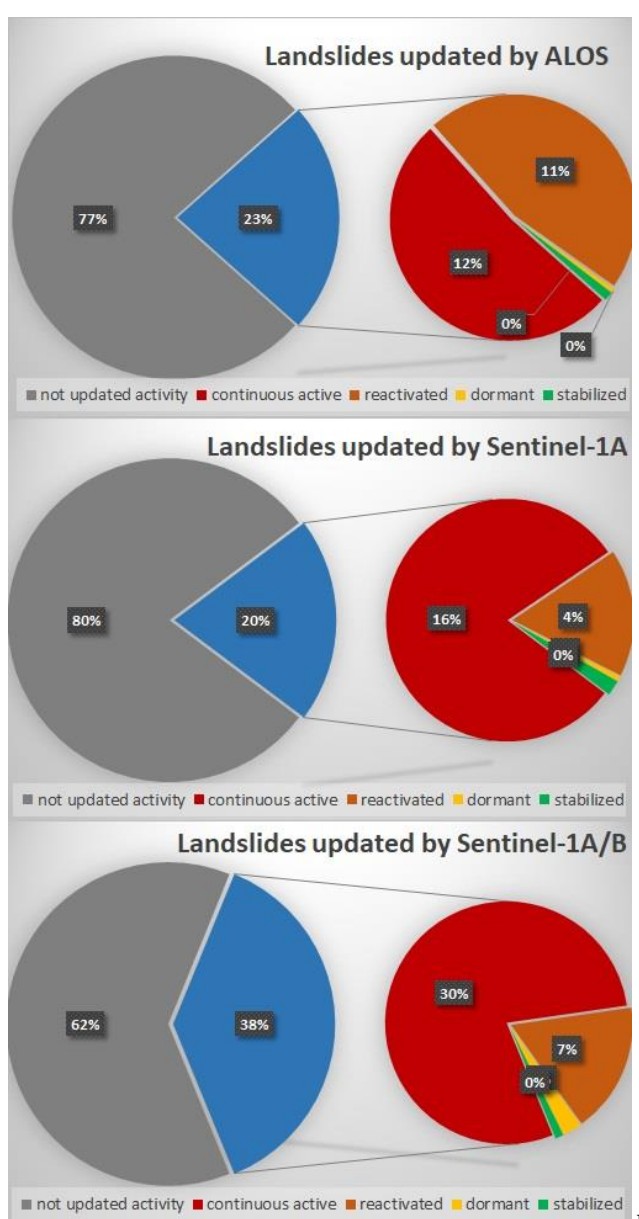

**Figure 9. Percentage of updated landslides**

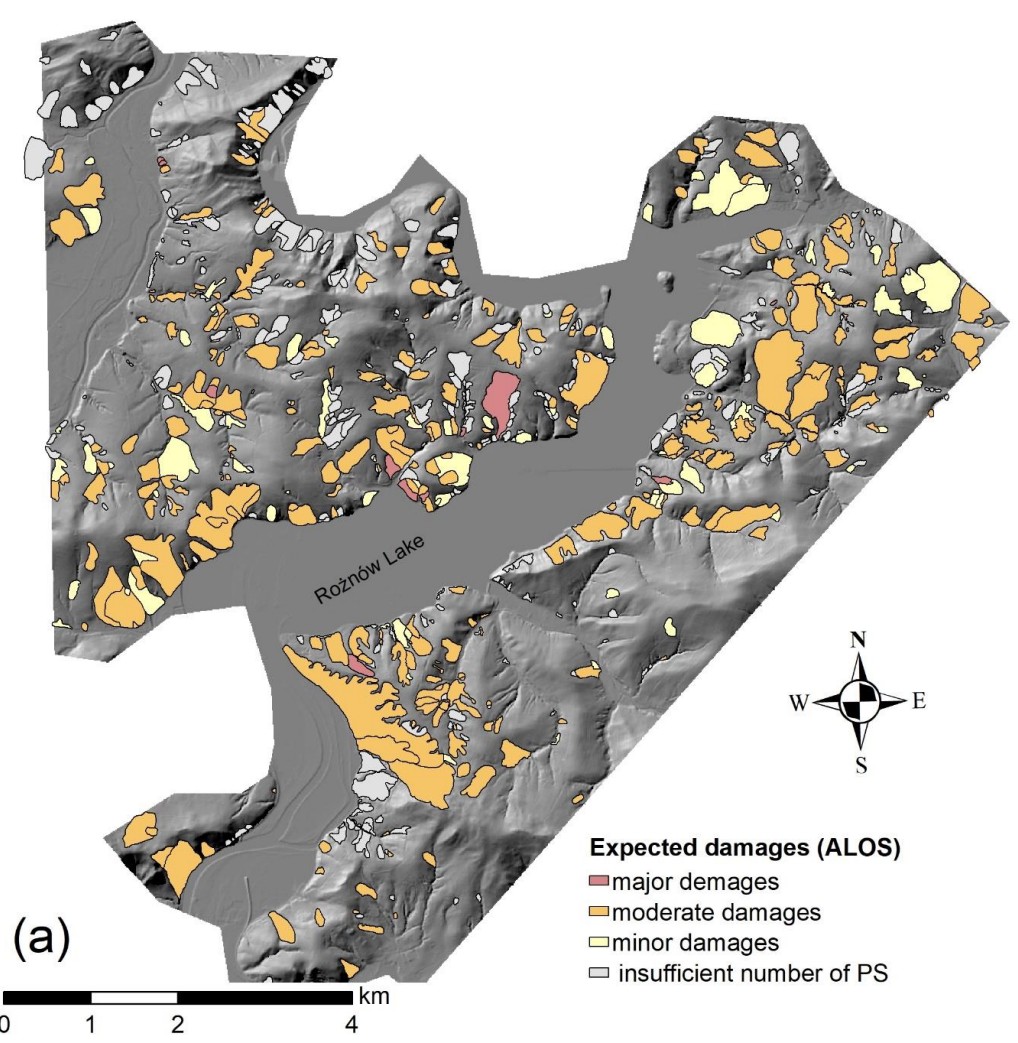

**Figure 10. Expected damages from landslides based on Mansour et al. (2011) and PSI results from ALOS (a), Sentinel 1A (b) and Sentinel 1A/B (c) data**
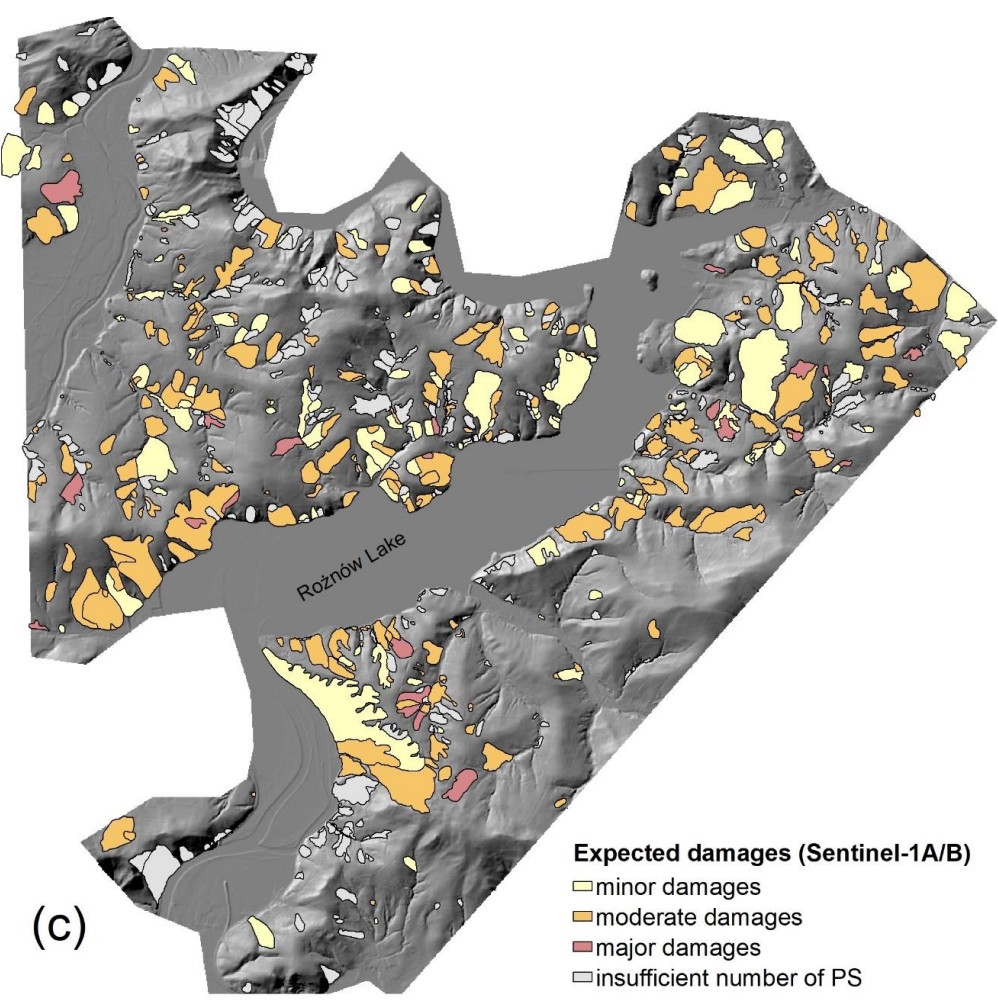

Expected damages (Sentinel-1A/B)
- minor damages
- moderate damages
- major damages
- insufficient number of PS

(c)

**Figure 10. Expected damages**
**from landslides based on Mansour et al. (2011) and PSI results from ALOS (a), Sentinel 1A (b) and Sentinel 1A/B (c) data**


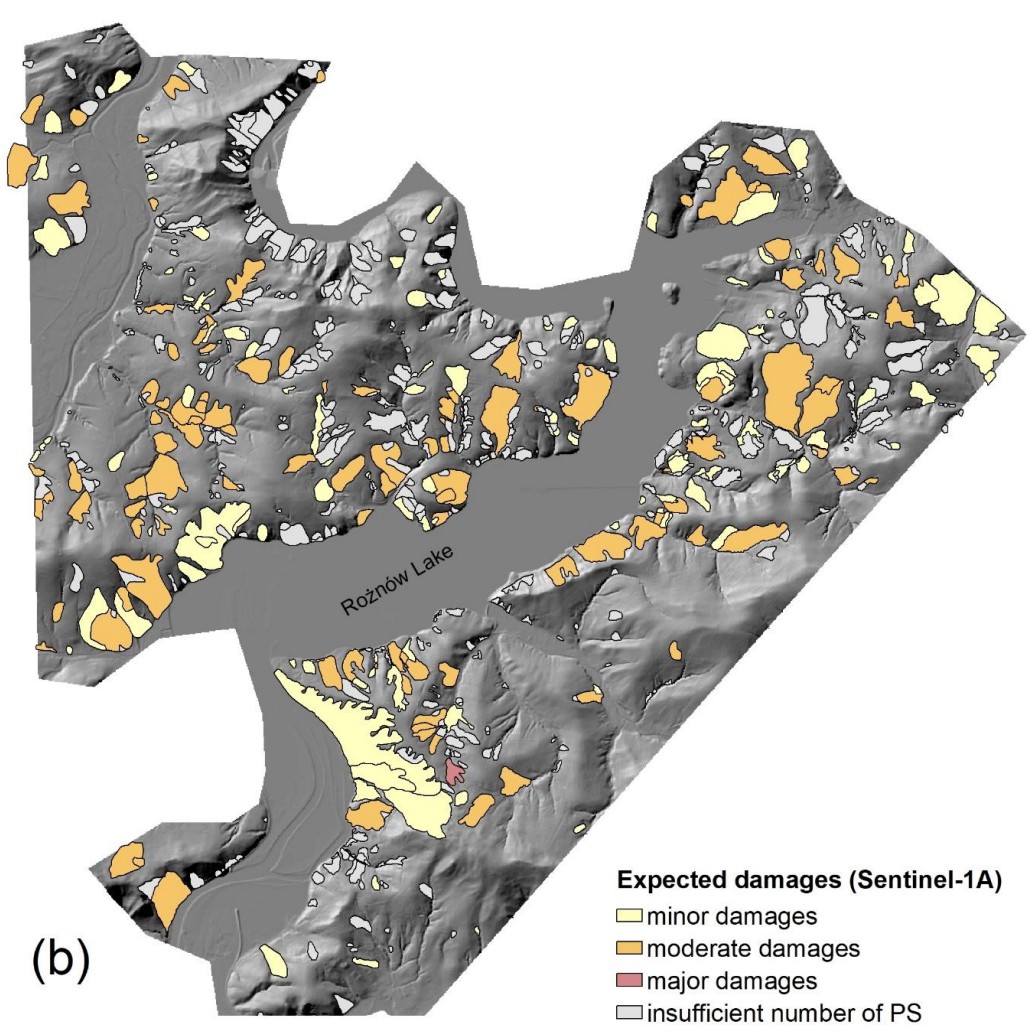

**Figure 10. Expected damages from landslides based on Mansour et al. (2011) and PSI results from ALOS (a), Sentinel 1A (b) and Sentinel 1A/B (c) data**

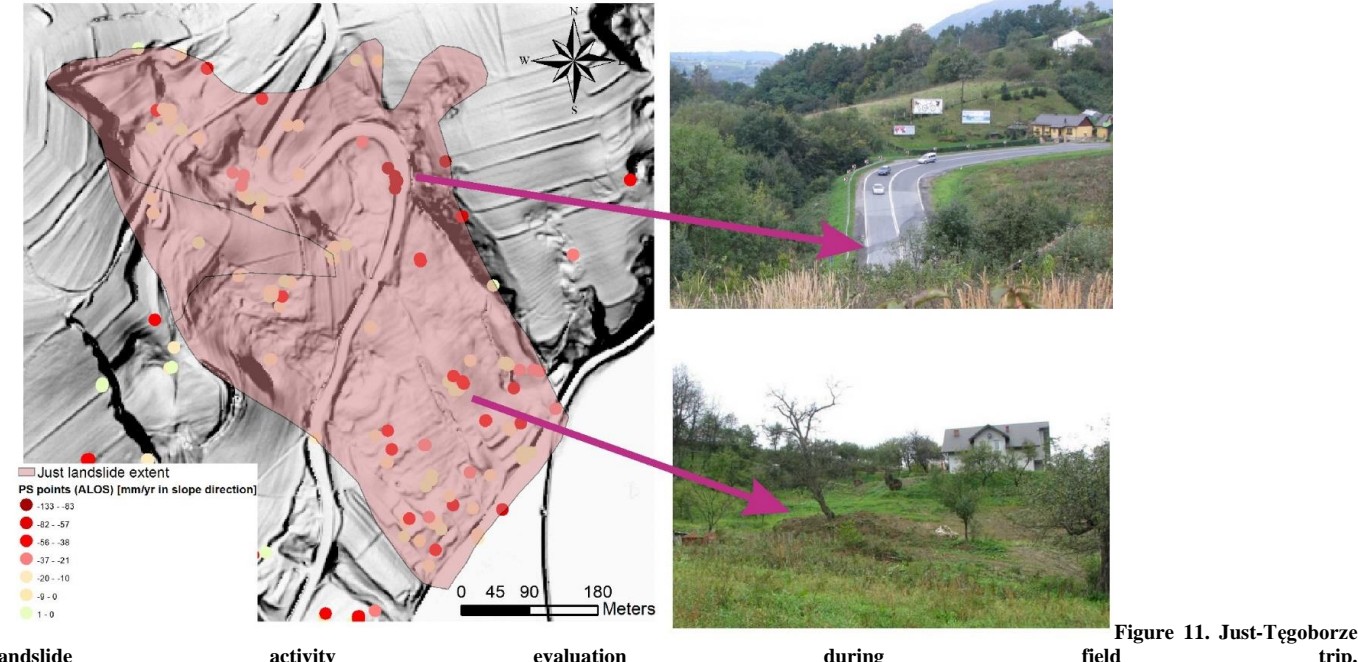

**landslide          activity          evaluation          during          field          trip.** Figure 11. Just-Tęgoborze



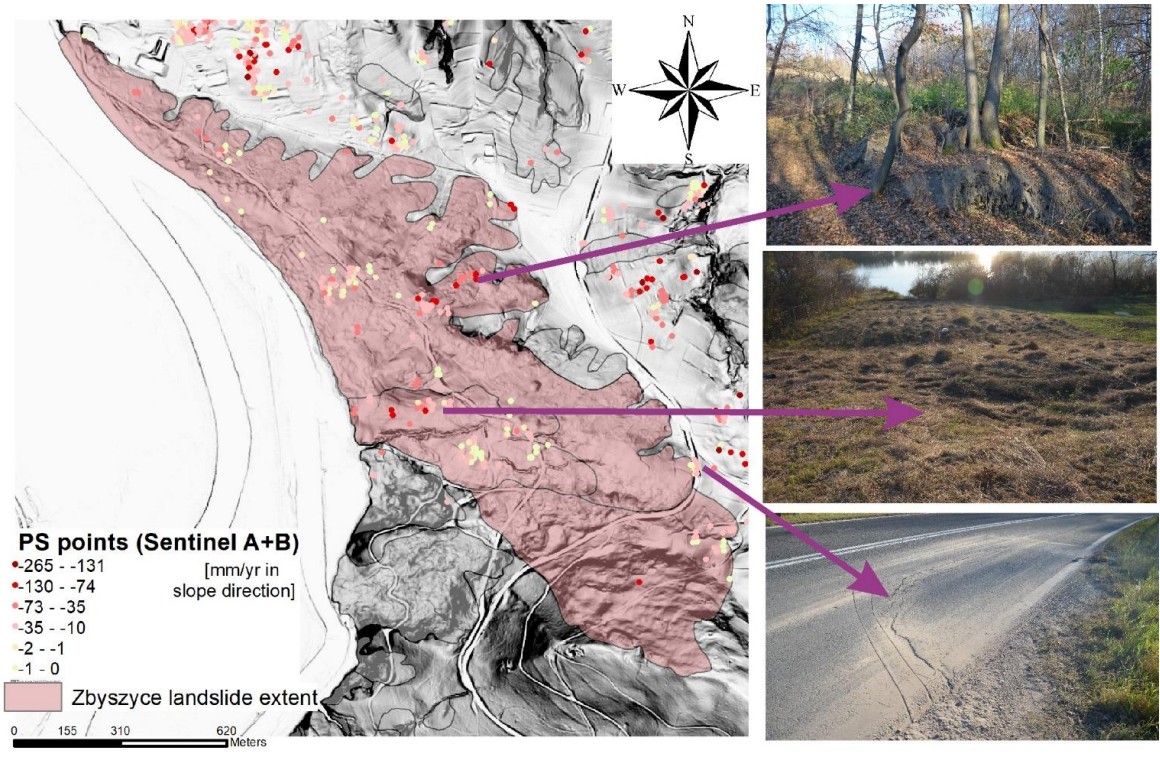

**Zbyszyce landslide activity evaluation during field** **Figure 12.** **trip.**




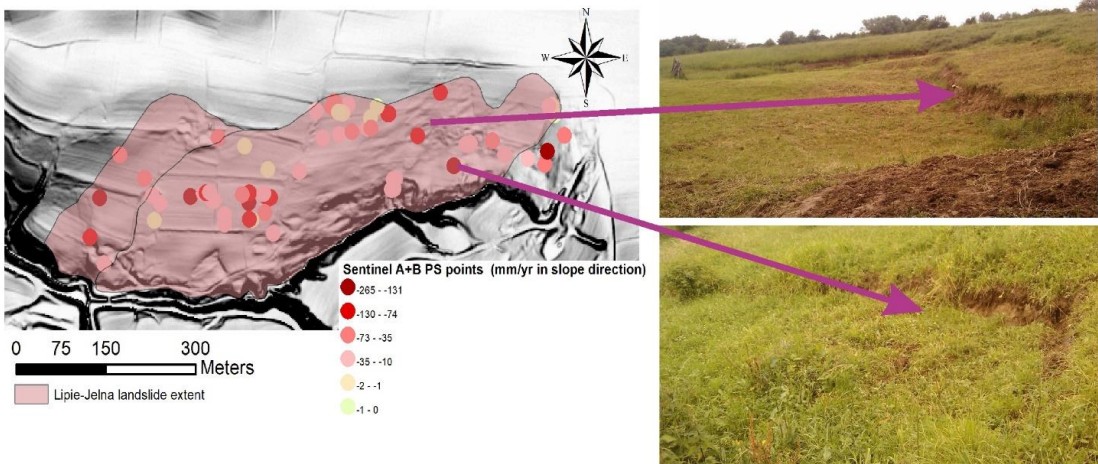

**Figure 13. Lipie-Jelna landslide activity evaluation during field trip.**


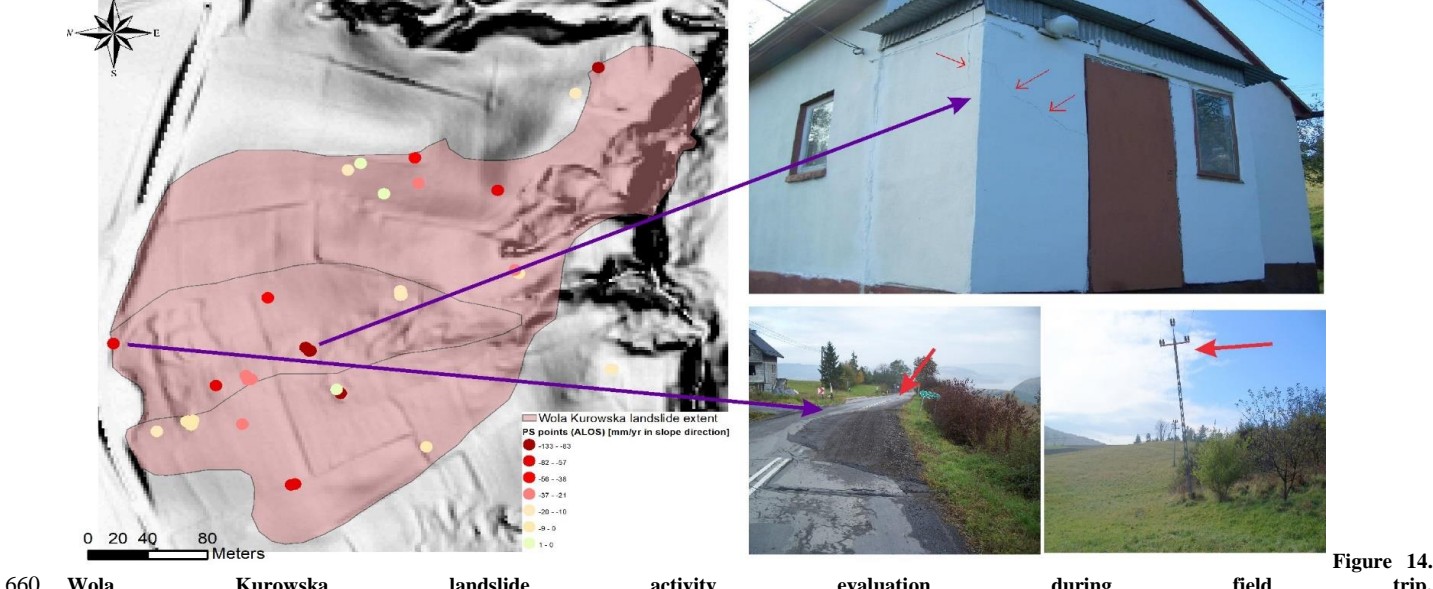

**Wola Kurowska landslide activity evaluation during field trip.**

**Figure 14.**


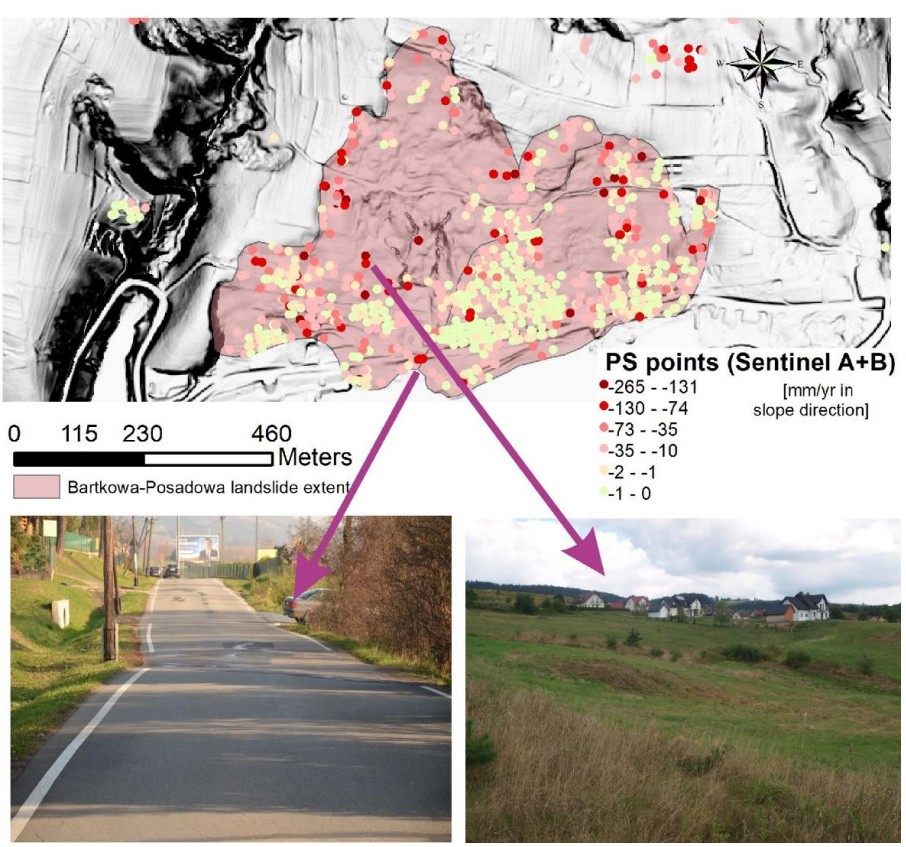

**Figure 15. Bartkowa-Posadowa landslide activity evaluation during field trip.**