# Peer review of "Multi-temporal landslide activity investigation by spaceborne SAR interferometry: Polish Carpathians case study"

_Natural Hazards and Earth System Sciences, 2020_

## Referee Comment (RC1) · Anonymous Referee #1 · 10 May 2020

The main scope of this work is to investigate the activity state of the landslides mapped in the 'SOPO' Database in the Malopolskie Municipality (Poland) area. The PSI approach is selected to measure displacements (Ferretti, et al. 2000) while the state of activity is obtained through a PSI-based matrix assessment (Cigna et al., 2013), and the potential damages to vulnerable elements using the method proposed in Mansour et al. (2011).

General Comments:

Despite a potential interest in some of the results of the analysis, the overall quality of the paper is well below the acceptable standards.

The paper is difficult to read: the English language is not of good quality; quite a few sentences are confounding and, here and there, the technical language is not precise or appropriate. The written would need a deep review. Furthermore, more care should have been put in the submission because paragraph numbering and figures numbering are both wrong.

The applied methods are used apparently without any, or sufficient, contextualization to the local characteristics, so, the assumptions are nor outlined properly, neither verified. How representative of the real displacement are the projections? How much does the DEM resolution impact in equations 1 and 2 and then on the final results? Is the minimum number of PSs landslide size-independent? landslide body part independent? No matter how PSs are clustered in the landslide body? Are the different velocity thresholds applied in the literature consistent with the local settings? The same for the expected damages.

Furthermore, I think that the conclusions are not supported by evidence because the 'field validation' does not seem to answer questions like the temporal relationship between damages and activity, but in particular, it is definitely biased by the analysis conducted only on one alleged class, and the analysis does not take enough seriously the impact of the assumptions and the meaning of getting different activity states according to the different used imagery. How to use 3 different (and uncombined) results?

Specific Comments:

Title

Is Multi-temporal related to investigation?

Abstract

5-10 "activity state verification of existing landslide inventory maps": it sounds like the state of activity of a map.

10-15

"overcome": perhaps mitigate.

"allows to homogenize the results from diverse acquisition modes and to compare displacement velocities": not always the displacements are along the steepest slope direction, this is an assumption.

15-20

'intensity': Is this a standard way to define the motion rate?

1 Introduction

20-25

'cumulative': in what sense?

'LIM': not all the inventories provide all the information.

'past and current landslides': what is the difference?

"Martha el al., 2010; Martha et al., 2012; Li et al., 2016": a bit surprised, these works are usually cited in relation to the (very good) results obtained using OBIA. I suggest to develop more the concept about the type of landslides that are tackled here. Slow-moving landslides usually need a stereoscopic view for being properly mapped.

30-35

'landslide detection': actually in most of the cases is the detection of displacements...May I also suggest to verify the use of DInSAR and InSAR here and there?

'limitations': for example?

45-50

'ambiguous': is it referred to the wrapped phase?

'it is challenging, if not impossible': I suggest to rephrase the sentence and say when

it is impossible.

'For example...': I suggest to better introduce the cited works.

'millimetre precision': referred to?

50-55

'these methods': I think this is correct and better than the list in 30-35, I suggest to try & merge the two.

'the state': of?

60-65

'for updating the landslide inventory': about the polygons or activity?

65-70

'specific thresholding.. are performed': I suggest to better explain this concept.

70 – 75

'applied': ?

75-80

'sensitivity to measure deformation over the study area': a 'collective' sensitivity? Not sure I understand this sentence correctly.

'difference in landslide activity updated': again, not sure I understand properly: different updates according to the different measures? But the activity is one... I guess.

1 Materials and Methods

2.1 Study area and existing landslide database

90-95

'around ... occurrence': there is a mix of 3 different concepts here: area affected by landslides, most active landslides, and frequency of landslide occurrence. Please, better sort out or connect all the pieces.

95-100

'seven times': what does it mean? 7 events? Seven maps?

100 – 105

'catastrophic landslide activity damaged': the term activity can generate confusion here, I suggest to change it.

2.1.1 Geological and hydrological settings of the study area

In my copy, the paragraph is repeated twice...

115 – 120

'10o to 35o': degrees

2.1.2 Landslide types and distribution

Again, in my copy, this paragraph is repeated twice.

130-135:

'among': in?

'hydro-geological conditions such as rock stratification and precipitation.': not sure I would label precipitation as a condition.

'Catastrophic landslide activity occurred in 2010': again, I suggest not to use here activity. I also suggest to better contextualize this sentence because these landslides cannot be monitored using DinSAR (and in most of the cases it does not make sense at all). Probably this sentence wants to introduce the following one but it should be better connected.

2.1.2 Pre-existing landslide inventory map

150-155

'the project': or the results of the project?

2 Methods

2-1 Radar data and PSI processing

170-175

'traditional': just to make it sure. Is this referred to as bi-temporal analysis or single interferograms? Anyway, if this is to justify the choice of the method, I suggest to move it in the discussion, together with the considerations related to the penetrating capacity of the X band, here only the method should be described.

2.2 PS post-processing phase

2.2.1 PS suitability analysis

2.2.2 PS velocity projection along the steepest slope

230-235

'Herrera et al. (2013)': they say: 'a minimum absolute value for cos $\beta$ = 0.3 is fixed for this study area'. Are the Authors sure that the same value can be used in a different area?

2.2.3 Velocity thresholding for activity state estimation – PSI based matrix approach

240-245

'Commonly, the average of LOS velocity estimates': I don't understand. Is it when more PSs are in a single landslide?

'Therefore, for activity state estimation, we applied 5 mm/yr as the Vslope threshold': I can't follow the reasoning. Did the Authors obtained the distributions and verified that

they were like in the citations?

Landslide intensity estimation

255-260

'landslides with sufficient information': when is the information sufficient? Is it only related to the PS availability?

3 Results

3.1 Landslide activity state and intensity map generation

270-275

'At least four PS points within a landslide body were set up as the threshold': the choice should be better justified. No matter what size, and other variables are? Is the distance among PSs taken into account? How were the different (at least 4) PS values combined to obtain Activity and Intensity maps?

290-295

'Therefore there are also landslides, which activity state has been updated based on historical or pre-existing data (SOPO database) if an insufficient number of PS were detected on the landslide object (compare also Fig. 5).': this step cannot be dumped this way... more details are needed.

3.2 Possible hazard assessment

General Comment: this is not a hazard assessment.

300-305

'Based on a literature review' conducted by the authors or by Mansour et al. (2011) )in which there is at least a try to characterize the vulnerable elements)?

305-310

'possible damages caused by mass movements in the study area are presented for three diverse PSI processing results.': so, how a decision-maker should read the maps? There are a few landslide areas that experience all the 3 classes, minor, moderate, and major damages. What should they expect?

3.3 Field validation

General comment: I can't understand how the list of the 'field verification' examples could verify the results. What I see in the figures are damages associated with some PSs, most of which with a velocity higher than 100mm /year. Shall I deduce the activity from the level of damage?

315-320

'confidence degree': what is it?

'measured displacements represent landslide dynamics': what does it mean? Landslide dynamic in the PS position? Or the entire landslide dynamic?

'reality assessment': ??

'moderate damage': why this choice? This for sure can create a bias in the investigation

'Activity states have been confirmed for 43 landslides': I guess the class, how did the expert confirmed the moderate class?

Landslide 'just-Tegoborze,' SOPO ID 23374

330 – 335

'Landslides tend to develop and increase activity over a large area.': the 23374 or in general?

Landslide 'Zbyszyce' SOPO ID 73253

Landslide 'Liie-Jelna' SOPO ID 73194

360 – 365

'When reactivated, this landslide covered about 80% of the landslide area': what does it mean?

Landslide 'Wola Kurowska' SOPO ID 73254

Landslide 'Bartkowa-Posadowa' SOPO ID 72917

4 Discussion

405-410

'landslide occurrence': Landslide occurrence or landslide monitoring? This sentence is a bad mix of different concepts.

'mostly connected': I suggest to relax the concept: 'can be largely connected...'

410 -415

'it was demonstrated that increasing the temporal sampling rate': I disagree, it was not formally demonstrated because the two series (12 and 6 days) don't overlap.

415 – 420

'the slope movement in a NE-SW direction represents only a small percentage of the real occurred displacements in LOS displacement rates...': This means that it is not only a matter of PS density. . . I think this point should have been taken into account more, both in the activity evaluation and in the discussion.

420-425

'homogenise landslide velocity': This is a (perhaps reasonable) assumption that needs to be verified. It can't come only from homogenization needs. . .

'real displacement rate can be much higher..': so how can a decision-maker trust and use this analysis?

430 – 435

'landslide activity states were evaluated by field': actually only a class was 'verified', so the evaluation is incomplete. I am not an expert in damages, so I can't say whether they were moderate or not.

'It can also support mentioned by Kroch..': I can't understand

Conclusions

---

## Referee Comment (RC2) · Anonymous Referee #2 · 14 May 2020

The authors have the data, but do not take a good advantage of them. In particular, they should show InSAR displacement time series to better demonstrate the state of activity of the studied landslides. What is new in the study with respect to many (tens or even hundreds) other similar studies that used InSAR for landslide assessment? Why is your study important? The authors closely follow some papers published several years ago, largely outdated. There is nothing new in that, unless the authors would like to focus on the problems and provide a critical assessment of these kind of simplistic approaches. However, this requires considerable landslide expertize and good knowledge of local slope processes, which, I'm afraid, the author may not have. So, I think, to bring out some novelty they need to re-focus on the application limitations and on some

potentially interesting aspects of the study (see below).

InSAR results look OK. However, the assumptions regarding their use need to be better justified, limitations clearly acknowledged. The use of some rather arbitrary "thresholds" appears improper. Therefore, some of the interpretations and conclusions are questionable. The manuscript is poorly organized, includes repetitions (for example, between part 1 and 2). Sloppy, even the sections and subsections of the manuscript are incorrectly numbered. Citations thrown in a haphazard fashion, often not representative or useful. Poor English.

Detailed comments

Abstract – should improve. What are the relevant scientific questions you are trying to address??

1 Introduction This part includes some information about PSI technique that should be moved to part 2 Materials and Methods

At line 51-72 In recent decades, new methods for updating landslide inventory maps – too long, too many and outdated references.

At line 65 Commonly used methodology in the abovementioned papers is the PSI matrix approach with diverse SAR sensors, where specific thresholding of the landslide velocity, acquired from specific PSI processing, are performed. – OK, but this belongs to Materials and Methods

Objectives – I suggest to focus more on the limitations and some potentially most interesting aspects of the study, that is: •Evaluating the effect of SAR geometry delivered from ascending and descending orbits from ALOS PALSAR and Sentinel 1 and the sensitivity to measure deformation over the study area; • Evaluating the difference in landslide activity updated from three diverse data stacks, namely: L-band (ALOS), Cband with one satellite (Sentinel 1A, with a revisit interval of 12 days) and C-band with two satellites (Sentinel 1A and 1B with a revisit time of 6 days), respectively; •

(New) Exploiting displacement time series for landslide activity/intensity assessment.

2 Materials and Methods This part mixes the description of the study area with the methods and includes unnecessary information like: This region is rural and, thanks to the breathtaking landscapes, the population has grown rapidly in recent decades. Moreover, it is also attractive to tourists.

At Line 106: landslide distribution within the study area as well their predefined activity states. – what do you mean by predefined?

2.1.1 Geological and hydrological settings of the study area No hydrological information is included, while different names of formations and units are thrown in and these are nowhere shown.

2.1.2 Landslide types and distribution (note that this section is repeated twice)

The landslide activity in the study area is mostly associated with hydro-geological conditions such as rock stratification and precipitation. These conditions created favorable conditions for landslide activation. – poorly written

Most of the landslide scarps within the study area lead down to valley floors – not clear

2.1 Radar data and PSI processing (page 6)

At line 177 Therefore, exploitation of C-band (Sentinel-1) and L- band (ALOS PALSAR) data can bring more advantages, especially in rural areas (Lu et al., 2018). – seems like an incorrect reference as the work is about land deformation in Changzhou city and uses InSAR data sets from 2006 to 2012, that is before Sentinel-1

2.2 PS post-processing phase (page 7)

After PSI processing, all results for the five diverse data sets have been post-processed in order to retrieve the most adequate displacement information – not clear what you mean by the most adequate, for what?

2.2.1 PS suitability analysis (page 7) – need to shorten as the information is already available in the literature

At line 204 Therefore, conversion of LOS deformation into the most probable direction (direction of maximum slope), by assuming a pure translational movement mechanism, is commonly used (Bianchini et al., 2012). – hopefully not so commonly, as, for example you also have rotational or combined rock-debris slides in your study area. This need to be explained.

2.2.2 PS velocity projection along the steepest slope (page 8)

At line 232 Despite the great advantage of the motion represented in the slope direction, this projection has some limitations. First, when $\beta$=90o, Vslope goes into infinity. Here we followed Herrera et al. (2013) and selected an absolute maximum value of $\beta$=72° - dangerous to talk about great advantage. Then, why should the approach by Herrera et al be good in your case?

2.2.3 Velocity thresholding for activity state estimation - PSI based matrix approach (page 8)

At line 247 For the LOS velocity, distribution is almost normal (Gaussian), while for SLOPE is second negatively skewed as a result of the PS reduction (Bianchini et al., 2013). Therefore, for activity state estimation, we applied 5 mm/yr as the Vslope threshold. – not very clear. Why 5mm/yr? What is your measurement precision?

At line 251 Four diverse activity states have been determined (Fig. 5): (1) reactivated = active after being inactive, (2) active continuous = currently moving, (3) dormant =inactive, but possible to be reactivated and (4) stabilised = not active anymore. – You have InSAR data and should show displacement time series to clearly demonstrate the state of activity of your landslides. For example, are they continuously moving?? I doubt. Don't some of them stop in a dry season or winter?

2.2.4 Landslide intensity estimation (page 9)

3. Results (page 9) 3.1 Landslide activity state and intensity map generation – part of this section belongs to the previous one (2 Methodology).

At line 271 However, the activity state has been presented only for landslides where sufficient PS points have been found. At least four PS points within a landslide body were set up as the threshold. – why just four PS? I know that some authors you cite have indicated and used (also their followers) this arbitrary threshold, but is it scientifically sound? You may not justify its use.

3.2. Possible hazard assessment (page 10) - part of this section belongs to the previous one (2 Methodology). However, why do you call it hazard assessment??? You are trying to assess possible damage. Moreover, please check the thresholds proposed by Mansour et al. (2011) really apply in your case and explain the limitations. At line 304 Landslide with velocity below 10mm we classified as landslide with minor expected damages. – questionable use of the threshold that can lead to dangerous interpretations. What if your landslide accelerates? And if t moves 9 mm/yr for 10 or more years?

3.3 Field validation (page 11) – again, part of this section belongs to the previous one (2 Methodology). However, what do you validate in the field? You make some inferences based only on some simple observations. Moreover, you when to the filed 10 years after the ALOS data were acquired, and 1-few years after the acquisition of Sentinel data. Need to explain.

Figures Fig 1 (and others) – lack of coordinates; incomplete caption

Fig 2 - > 60 geological units with no explanation, sending the reader to a supplementary material? You need to group them by similar lithology, with reference to the susceptibility to landsliding

Figs 4-6 Who needs to see these figures?

Fig 6 – Present or historical PSI data? Not clear what you mean.

Fig 7 – overlapping colors, hard to see what is going on.

Fig 8 – Landslide intensity scale or simply velocity scale?

Fig 11 – wrong choice of color for the landslide area, as some PS have the same or very similar color

---

## Referee Comment (RC3) · Anonymous Referee #3 · 22 May 2020

The main goal of this manuscript is to present the updated state of activity of landslides in Małopolskie municipality, a rural area with sparse urbanization in the Polish Flysch Carpathians, based on InSAR results. At a glance the manuscript seems to be a good case study, carried out according to state-of-the-art methodologies published and highly cited in the specialist literature. However, after a careful reading, it comes out that there is no element of novelty in both the InSAR processing & post-processing methodology and contribution to advance the field of landslide studies. The area where this manuscript could bring in a novel contribution could be the improved knowledge about local landslides in Małopolskie and the risk that they pose to population, buildings and infrastructure. However, despite the field validation, it is difficult to relate the

conclusions achieved by the authors based on InSAR processing and analysis with the challenges that this area in the Polish Carpathians is experiencing.

So in general I believe that this manuscript (and the research behind it) requires more work to be publishable.

Further detailed comments are reported below.

Abstract It can be improved by removing redundancies (e.g. lines 12-13 vs 18), making explicit the cause for 7 landslides out of the total 50 that could not be confirmed, and highlighting the novel contribution with regard to either methodological approach or knowledge about local landslide issues and the risks that they pose (see also what the authors state at lines 84-87).

Introduction The strength of this section is for sure the wide and comprehensive literature review. Key and relevant papers are cited. However, although at line 72 the authors start listing the objectives of the manuscript, it is not clear how these objectives relate to the literature reviewed in the previous paragraphs. In which way is this manuscript novel (if it is) compared to the cited literature? The feeling is that the authors primarily wanted to ensure that their methodology was aligned with the literature.

Figure 1: it is not clear whether the landslides mapped are those from the pre-existing inventory, i.e. prior to the update based on InSAR. The caption should be more explicative

Methods The methodology is in general correct because it largely relies on well established and accepted methodologies. However, it is difficult to see it framed within the specific geological, geomorphological and environmental features of the study area landscape. Therefore, although they may be correct, the rationale for the implementation of some of the assumptions is difficult to understand. At what extent the knowledge of the local landslides helped the authors to shape and adapt the methodology? With regard to PSI, some key information are missing in the text (or I was not able to find

myself), e.g. the location of the reference points selected during the processing.

Figure 3: there is a typo in Feretti et al.; it should be Ferretti et al.

Furthermore, in the text I did not find the explanation of the rationale followed by the authors for processing first Sentinel-1A data only and then Sentinel-1A + Sentinel-1B data together. Would not it be better to process directly Sentinel-1A + Sentinel-1B data together? What is the advantage of this approach? More in general, what is the impact of each satellite dataset on the state of activity?

Discussion I would have expected more linkage between the evidence gathered during the field validation and the InSAR data. When did the damages observed in the field happen? Is there a temporal association/correlation between the displacement showed by InSAR data and the damages?

Given the current statements at lines 432-436, it is not clear how the authors assessed the degree of damage to buildings and infrastructure.

Several minor typos throughout the text need to be corrected, e.g. - line 16: "filed" to be corrected into "field" - line 181: "Feretti et al." to be corrected into "Ferretti et al."

---

## Author Comment (AC1) · 24 Jul 2020

The authors are very grateful to the Editors and Associate Editors for the kind consideration and possible publication of our article in the Natural Hazards and Earth System Sciences. The authors would like to thank all reviewers for suggesting improvements for the manuscript. Point-wise reply/answer to each comment is provided below (comments are shown in **green bold font**, answers are shown as black font). All suggestions have been addressed, but still, the authors are open for further explanations and cooperation in term of manuscript improvement. Furthermore, the authors appreciate the editors and reviewers for the timely handling of the review process. After reviewer's evaluation, we have the clear idea how to correct and improve our manuscript to be publishable in NHESS. This mainly includes:

- *Rewriting the introduction to highlight the novel aspect of the manuscript because in the present form, the novelty of the work is not clearly visible (rewriting according to the reviewer 2 suggestion)*
- *The literature review will be moved into material and methods sections (according to the reviewers 2 suggestion, shorten a little however, this literature review will be kept because as the reviewer 3 stated this is a strong part of the introduction)*
- *According to reviewer 2 suggestion, time series of the PS points will be presented together with the discussion about presented deformation mechanism*
- *According to reviewer 1, some sentences will be rewritten to improve readably and some confusing aspect (thresholding, slope reprojection, PS technique limitation etc.) will be deeply discussed.*

*Authors*

**REVIEWER 1**

**The main scope of this work is to investigate the activity state of the landslides mapped in the 'SOPO' Database in the Malopolskie Municipality (Poland) area. The PSI approach is selected to measure displacements (Ferretti, et al. 2000) while the state of activity is obtained through a PSI-based matrix assessment (Cigna et al., 2013), and the potential damages to vulnerable elements using the method proposed in Mansour et al. (2011).**
**General Comments:**
**Despite a potential interest in some of the results of the analysis, the overall quality of the paper is well below the acceptable standards.**
**The paper is difficult to read: the English language is not of good quality; quite a few sentences are confounding and, here and there, the technical language is not precise or appropriate. The written would need a deep review.**

Actually, the manuscript has been checked by the native speaker before submission. Nevertheless, we will try to improve the English and we will send it again for another proof-check and we will attach the certificate.

**Furthermore, more care should have been put in the submission because paragraph numbering and figures numbering are both wrong.**

It was an unfortunate when preparing the manuscript according to the Copernicus NHESS template. This will be of course corrected.

**The applied methods are used apparently without any, or sufficient, contextualization to the local characteristics, so, the assumptions are nor outlined properly, neither verified.**

*The applied methodology is widely used in the literature in various papers. Below we provide works which used similar approach as we used (similar because not all authors investigated all aspect → RI index, Cindex, slope reprojection etc)*

**PSI based matrix for activity state:**

*PSI based matrix using Envisat data → Del Ventisette et al. (2014); Bianchini et al. (2013)*

*Threshold of $v_{LOS}$= -2mm  used in → Del Ventisette et al. (2014) and threshold of $v\_slope = -5mm$ used in Bianchini et al. (2013)*

**PSI-based matrix for intensity scale:**

*In general, tree various thresholds exist in literature to distinguish slow moving and extremely slow moving landslide.*

*Righini et al.2011 used $v_{LOS}$=-10mm to distinguish slow up to extremely slow mowing landslides*

*Bianchini et al. (2012) used $v_{LOS}$=10mm  to distinguish slow up to extremely slow mowing landslides*

*Cigna et al. (2013) used $v_{slope}$ =13mm to distinguish slow up to extremely slow mowing landslides*

*Kalia (2018) used $v_{slope}$ =16mm to distinguish slow up to extremely slow mowing landslides*

*Cruden and Varnes (1996) recommended to used 16mm to distinguish between extremely slow landslide and slow landslide. Extremely slow landslides have velocity smaller than 16 mm/yr and very slow landslides have velocity between 16 mm/yr and 1.6 m/yr),*

***How representative of the real displacement are the projections?***

*Projections are limited by the DTM accuracy, that is pretty good because we use LiDAR DTM. Slope velocities are higher than LOS displacement. This reprojection has been used by Cigna et al. (2013) and Kalia (2018).Deformation in slope direction are generally mathematical scaling of LOS deformation. Below, we provided the picture, which represents reprojections representation from Béjar-Pizarro et al.2017.*

[Figure]

[Figure]

*Source: Béjar-Pizarro, M., Notti, D., Mateos, R. M., Ezquerro, P., Centolanza, G., Herrera, G., ... & Fernández, J. (2017). Mapping vulnerable urban areas affected by slow-moving landslides using Sentinel-1 InSAR data. Remote Sensing, 9(9), 876.*

***How much does the DEM resolution impact in equations 1 and 2 and then on the final results?***

*It depends on the length of the slope. For long slopes with almost constant slope angle, the DEM resolution can be lower. However, this issue seems to be not investigated in the literature yet, but it is possible to assess this impact by means of the error propagation rule. Appropriate calculations will be provided in the improved version of the manuscript. Thank you for this remark.*

***Is the minimum number of PSs landslide size-independent? landslide body part independent? No matter how PSs are clustered in the landslide body?***

*To some extent yes, because if a specific landslide is the complex one this landslide is divided into the smaller parts homogeneous in terms of morphology, type of the movement etc. (this was made by geologists in the field while creating the national landslide inventory map). Therefore, we would like to be consistent and we update the activity state of this specific homogeneous landslide parts. We didn't used any clustering etc. because activity state is presented for the whole landslide body rather than some specific part of it, thus we are interested in general if something is moving inside the landslide body. A similar approach (using 4 points within landslide body) has been utilized by Bianchini et al. (2013); Cigna et al. (2013). In literature, there are also examples of even 3 points within landslide body → Cascini et al. (2013)*

***Are the different velocity thresholds applied in the literature consistent with the local settings? The same for the expected damages.***

*We do not understand what is meant "local setting". We considered speed of the landslide movement in order to correctly use presented method. Parameters of the processing are not adjusted to the specific local setting because due to the persistent scatterers limitation PSI-based matrix approach can be used only for small and very small moving landslide. Therefore, we have investigated landslide documentation over the study area, to be sure that PSI based matrix approach can been used for landslide activity assessment in this study area. The same was stated by other authors that this methodology cannot be applied for fast and rapid landslide. We will give more clarification connected with this issue when revising the manuscript.*

***Furthermore, I think that the conclusions are not supported by evidence because the 'field validation' does not seem to answer questions like the temporal relationship between damages and activity,***

*Damages in presented study was used for the evaluation/prove of the activity state (active, not active) because buildings and infrastructures damages are a direct sign/evidence of ground movement. When some infrastructure/building damage exists within the landslide body we can directly conclude that this landslide is active. Of course, this field investigation has been performed for specific time. This means that for each of the multi-temporal map (ALOS/SENTINEL-1A/SENTINEL-1A/1B), three various field campaigns have been performed For ALOS-> year 2010, for Sentinel1a-> each 2016 and for Sentinel1a/b ->year 2017.*

***but in particular, it is definitely biased by the analysis conducted only on one alleged class, and the analysis does not take enough seriously the impact of the assumptions and the meaning of getting different activity states according to the different used imagery.***

*We investigated active and not active states, but we presented the photos and some specific example for active landslide only. If there are no evidences (damage) for not active (dormant/stabilized) landslide then there is no point to present the photos taken in field.*

***How to use 3 different (and uncombined) results?***

*These 3 different results are used for multi-temporal landslide activity state generation. The multi-temporal means that when we have landslide database in any GIS software, we have landslide object and we can check the attributes connected with activity for various year. For instance, we can see that landslide X was active in 2010 then in 2016 not active and in 2017 again active. This will facilitate another studies for instance evaluation of the precipitation threshold/amount which makes this landslide active. This is especially important aspect in Polish Carpathians because this is the main driving force of the landslide activity in this specific region.*

***Specific Comments:***

***Title Is Multi-temporal related to investigation?***

*Yes, because we investigated various time spans:*

*For ALOS it is 31/01/2008-27/12/2010*

*For sentinel-1A it is 29/11/2014-24/12/2016*

*For sentinel-1A/B it is 2/01/2017-31/12/2017*

*Abstract*

*5-10 "activity state verification of existing landslide inventory maps": it sounds like the state of activity of a map.*

*For clarification, we investigated landslide activity state. If this sentence is misleading, we will correct it when revising the manuscript.*

*10-15"overcome": perhaps mitigate.*

*Yes, this word can be changed as you suggested, when revising manuscript.*

*"allows to homogenize the results from diverse acquisition modes and to compare displacement velocities": not always the displacements are along the steepest slope direction, this is an assumption.*

*Yes, but we reprojected them into the steepest slope in order to have ascending and descending results in one database, because using LOS we cannot combine results from ascending and descending geometry.*

*15-20 'intensity': Is this a standard way to define the motion rate?*

*Yes, we use standard nomenclature, e.g.*

*Cigna, F., Bianchini, S., Casagli, N. 2013. How to assess landslide activity and intensity with Persistent Scatterer Interferometry (PSI): the PSI-based matrix approach. Landslides, 10(3), 267-283. https://doi.org/10.1007/s10346-012-0335-7.*

*1 Introduction*

*20-25 'cumulative': in what sense?*

*This word will be removed when revising the manuscript.*

*'LIM': not all the inventories provide all the information.*

*That's true, it depends on the country. In Poland, we can find various information in landslide inventory database. This issue will be addressed in the revisited version of the manuscript.*

*'past and current landslides': what is the difference?*

*It is desirable to systematically update landslide databases. Especially, this should be made after each catastrophic event (heavy precipitations) when a lot of new landslide were activated. In Poland, we had two such important catastrophic events: in 2001 and in 2010. We will think how to rewrite this sentence to be more clear.*

*"Martha el al., 2010; Martha et al., 2012; Li et al., 2016": a bit surprised, these works are usually cited in relation to the (very good) results obtained using OBIA.*

*Yes, you are right. But this object-oriented approach is used for optical images therefore these works can also be used for citations related to landslide mapping using satellite images.*

*I suggest to develop more the concept about the type of landslides that are tackled here.*

*Slow-moving landslides usually need a stereoscopic view for being properly mapped.*

*Information about the landslide type is presented in section 2.1. However, we can try to add some information here.*

*30-35 landslide detection': actually in most of the cases is the detection of displacements: : :May I also suggest to verify the use of DInSAR and InSAR here and there?*

*'limitations': for example?*

*We have InSAR, DInSAR and PSInSAR. PSInSAR is a specific modification of DInSAR, which minimizes atmospheric artefacts. DInSAR is similarly specific modification of InSAR, which is used for deformation estimation. Since landslides located in our study area are slow and extremely slow moving landslide, the movement of few mm/y will be not visible in DInSAR results. This is mostly*

because of atmospheric artefacts and another DInSAR limitation. PSInSAR thanks to its possibility of atmospheric phase screen modeling, has the ability to increase accuracy of deformation estimates.

***45-50 'ambiguous': is it referred to the wrapped phase?***

*Yes*

***'it is challenging, if not impossible': I suggest to rephrase the sentence and say when it is impossible.***

*We will add this information when revising the manuscript*

***'For example...': I suggest to better introduce the cited works.***

*We will correct it*

***'millimetre precision': referred to?***

*Referred to PSI technique, we will add this information in the revised version.*

***50-55 'these methods': I think this is correct and better than the list in 30-35, I suggest to try & merge the two.***

*We will try to merge these two into the one section, however another reviewer suggests to move literature review into the method section. Therefore, we will try to only present here the possibility and limitation of InSAR technique for monitoring of landslides.*

***'the state': of?***

*Activity state, the word of "activity" will be added*

***60-65 'for updating the landslide inventory': about the polygons or activity?***

*About the activity not boundary. We will clarify this.*

***65-70 'specific thresholding.. are performed': I suggest to better explain this concept.***

*We will add, info about thresholds used by various authors and we will provide short discussion.*

1) *For landslide activity:*

   $v_{LOS}$= -1.5mm Envisat, → Del Ventisette et al. (2014)

   $v_{LOS}$= -2mm,Envisat , →Bianchini et al. (2013)

   $v_{slope}$ = -5mm, Envisat , →Bianchini et al. (2013)

   $v_{slope}$ = -5mm, ERS, Radarsat 1&2, → Cigna et al. (2013)

   $v_{slope}$ = -10mm, Sentinel-1 → Kalia (2018)

2) *For various intensity. In general, tree various threshold exists in literature to distinguish slow moving and extremely slow moving landslide.*

*Righini et al.2011 used $v_{LOS}$=-10mm,*
*Bianchini et al. (2012) used $v_{LOS}$=10mm*
*Cigna et al. (2013) used $v_{slope}$ =13mm.*
*Kalia (2018) used $v_{slope}$ =16mm.*

*As can be noticed, higher threshold are used in case of $v_{slope}$ (velocity reprojected into the steepest slope). In our study, we decided to used $v_{slope}$ =16mm similarly to Kalia (2018) but the most important reason is that this threshold is presented in well-know, old and widely respected literature (4218 citations) of Cruden and Varnes (1996). They stated that extremely slow landslide has velocity < 16 mm/yr and very slow landslides (16 mm/yr < velocity < 1.6 m/yr), Having considered abovementioned issues, we decided to applied threshold of $v_{slope}$ =16mm.*

***70 – 75 'applied': ?***

*We will change the word into "used" or "utilized"*

***75-80 'sensitivity to measure deformation over the study area': a 'collective' sensitivity? Not sure I understand this sentence correctly.***

*"sensitivity to measure deformation over the study area". Since the radar technique is not appropriate in each case (it depends on the sensor geometry and terrain slope and aspect). Thus we wanted to assess if specific sensor can be used for landslide monitoring in specific regions. We will try to clarify this issue when revising our manuscript.*

**'difference in landslide activity updated': again, not sure I understand properly: different updates according to the different measures? But the activity is one… I guess.**

*Activity state. It means active, dormant stabilized etc. we will add this information for more clarification.*

**1 Materials and Methods**

**2.1 Study area and existing landslide database**

**90-95 'around … occurrence': there is a mix of 3 different concepts here: area affected by landslides, most active landslides, and frequency of landslide occurrence. Please, better sort out or connect all the pieces.**

*We will try to rewrite it as area mostly affected by landslides*

*"The region is known for its frequent landslide occurrence." --> we will rewrite it as region is known for its landslide density and serious damages caused by landslides*

**95-100 'seven times': what does it mean? 7 events? Seven maps?**

*Seven times means 7 catastrophic events, such as mentioned before about the precipitation. Generally, heavy precipitation "can create" new landside in the study area. We will try to rewrite it in revised version.*

**100 – 105 'catastrophic landslide activity damaged': the term activity can generate confusion here, I suggest to change it.**

*It will be rewritten as "Recently, landslides have caused catastrophic damage due to abundant rainfall and flooding in 1997 and 2010."*

**2.1.1 Geological and hydrological settings of the study area**

**In my copy, the paragraph is repeated twice...**

*Unfortunately, when coping our plain text into the Copernicus NHESS template we copied this twice. It will be corrected of course.*

**115 – 120 '10o to 35o': degrees**

*Degree symbol will be used*

**2.1.2 Landslide types and distribution**

**Again, in my copy, this paragraph is repeated twice.**

*Unfortunately, when coping our plain text into the Copernicus NHESS template, we copied this twice. It will be corrected.*

**130-135: 'among': in?**

*The word "among" will be changed into "in the" or "within"*

**'hydro-geological conditions such as rock stratification and precipitation.': not sure I would label precipitation as a condition.**

*We will try to label precipitation with another word rather than condition*

**'Catastrophic landslide activity occurred in 2010': again, I suggest not to use here activity. I also suggest to better contextualize this sentence because these landslides cannot be monitored using DinSAR (and in most of the cases it does not make sense at all). Probably this sentence wants to introduce the following one but it should be better connected.**

The direct driving force of landslides in this region is precipitation. When abundant rainfalls occurred, then a lot of landslide started to be active. Thus, we are using this word. We will try to be more clear in the revisited version of the manuscript.

*2.1.2 Pre-existing landslide inventory map*
*150-155 'the project': or the results of the project?*

We think that there is not wrong to use the project if it means that this is the goal of the project. But we can use also the results of the project.

*2 Methods*
*2-1 Radar data and PSI processing*
*170-175 'traditional': just to make it sure. Is this referred to as bi-temporal analysis or single interferograms?*

Yes, by using 'traditional' we mean single interferograms, not advanced methods based on interferogram stacking like PS or SBAS

*Anyway, if this is to justify the choice of the method, I suggest to move it in the discussion, together with the considerations related to the penetrating capacity of the X band, here only the method should be described.*

This can be moved into the discussion section when revising our manuscript

*2.2 PS post-processing phase*
*2.2.1 PS suitability analysis*
*2.2.2 PS velocity projection along the steepest slope*
*230-235 'Herrera et al. (2013)': they say: 'a minimum absolute value for cos β = 0.3 is fixed for this study area'. Are the Authors sure that the same value can be used in a different area?*

Yes, because these thresholds do not depend on the study area bur rather it depends on geometrical limitations. This approach was already applied by Bianchini et al. (2013) Kalia (2018). The C coefficient is the fraction of the 3D displacement that can be measured by PS targets and $\beta$ the angle between the steepest slope and the LOS direction. Several limitations of this method must be taken into account: when $\beta$ is almost 90°, C is close to 0 and $V_{SLOPE}$ tends to infinity. Following the work done by Herrera et al. 2013, an absolute maximum value of $\beta = 72°$ corresponding to cos $\beta = 0.3$ is fixed and, as a result, $V_{SLOPE}$ cannot be higher than 3.33 times the $V_{LOS}$. This threshold corresponds to the condition number of 15 proposed by Cascini et al. 2010 as the number for the inversion matrix solving the algebraic system used for the projection process. In order to reduce any $V_{SLOPE}$ exaggeration, it is assumed C = −0.3 when −0.3 < C< 0 and C = 0.3 when 0 < C < 0.3. Whereas PS $V_{SLOPE}$ values turn positive ($V_{SLOPE}$ > 0) they are discarded. This is because the positive $V_{SLOPE}$ would represent uphill movement.

*2.2.3 Velocity thresholding for activity state estimation – PSI based matrix approach*
*240-245 'Commonly, the average of LOS velocity estimates': I don't understand. Is it when more PSs are in a single landslide?*

Yes, the average of at least 4 points have been used.

Based on the literature review, we decided to used 4 points, similarly to other authors Bianchini et al. (2013); Cigna et al. (2013). Also the average of these at least 4 points has been used similarly to authors of Bianchini et al. (2012 ); Bianchini et al. (2013); Cigna et al. (2013).

In the revised version we will implement the following improvement. Based on at least 4 points we will calculate the trimmed mean, removing two values; the smallest and the largest one. This will allow us to reduce outliers influence.

*'Therefore, for activity state estimation, we applied 5 mm/yr as the Vslope threshold': I*

*can't follow the reasoning. Did the Authors obtained the distributions and verified that they were like in the citations?*

They were like in the citations.

$v_{LOS}$= -1.5mm Envisat, → Del Ventisette et al. (2014)

$v_{LOS}$= -2mm,Envisat , →Bianchini et al. (2013)

$v_{slope}$ = -5mm, Envisat , →Bianchini et al. (2013)

$v_{slope}$ = -5mm, ERS, Radarsat 1&2, → Cigna et al. (2013)

$v_{slope}$ = -10mm, Sentinel-1 → Kalia (2018)

Since for $v_{slope}$ = -5mm is usually applied, we decided to use this threshold.

**Landslide intensity estimation**

**255-260**

**'landslides with sufficient information': when is the information sufficient? Is it only related to the PS availability?**

Yes, this means, if at least 4 points inside the landslide body is detected (after removing these with high C value), then the activity state could be evaluated for his specific landslide. This has been written in the manuscript as *"At least four PS points within a landslide body were set up as the threshold" (lines 273-274)*

**3 Results**

**3.1 Landslide activity state and intensity map generation**

**270-275 'At least four PS points within a landslide body were set up as the threshold': the choice should be better justified. No matter what size, and other variables are? Is the distance among PSs taken into account? How were the different (at least 4) PS values combined to obtain Activity and Intensity maps?**

As we previously mentioned ,we followed the procedure presented by other authors in Bianchini et al. (2012 ); Bianchini et al. (2013); Cigna et al. (2013) with small modification. Complex landslides have been divided into the specific part based on their morphology. I was made officially by geologist in the field. We applied the procedure used to particular landslide parts. Thus we add the information in text as". *A number of landslides in the study area are complex landslides. Such landslides are divided into parts and represented in SOPO databases as separate objects. In this case, the mentioned threshold is related to each landslide object."* However, we will try to clarify this while revising the manuscript.

**290-295 'Therefore there are also landslides, which activity state has been updated based on historical or pre-existing data (SOPO database) if an insufficient number of PS were detected on the landslide object (compare also Fig. 5).': this step cannot be dumped this way… more details are needed.**

As previously mentioned, we will clarify this.

**3.2 Possible hazard assessment**

**General Comment: this is not a hazard assessment.**

Ok, according to Mansour et al. (2011) it is called: „Expected damage from displacement." Thus, we label this subsection as „Expected damages."

**300-305 'Based on a literature review' conducted by the authors or by Mansour et al. (2011) ) in which there is at least a try to characterize the vulnerable elements)?**

Based on the authors' literature review. We will clarify this aspect.

**305-310 'possible damages caused by mass movements in the study area are presented for three diverse PSI processing results.': so, how a decision-maker should read the maps? There are a few landslide areas that experience all the 3 classes, minor, moderate, and major damages. What should they expect?**

This tree various maps are muli-temporal. More specifically, estimated for various time spans. Thus, the most recent one is generated for sentinel-1a and 1b data. However, old maps for instance generated based on ALOS data was used to show the past landslide activity states and possible damage generation. Therefore, in the subsection connected with field validation, we show specific landslides with velocities and activity estimated for instance for ALOS for the time span 31/01/2008-27/12/2010 and photos taken at 9/29/2010. Based on past demonstration, we can see the relationship between PS velocities (activity state) and real damages.

*3.3 Field validation*

*General comment: I can't understand how the list of the 'field verification' examples could verify the results. What I see in the figures are damages associated with some PSs, most of which with a velocity higher than 100mm /year. Shall I deduce the activity from the level of damage?*

Yes, you can deduce that this is active landslide assessed based on PS. However in the manuscript, we described multi-temporal activity state for each landslide for instance: *"This landslide was assessed as active (in 2010), dormant (2014-2016) and reactivated (in 2017) based on the PSI-based matrix approach. There are expected moderate damages in 2010 and 2017 and minor damages in 2016 due to landslide activity."* Thus, one don't need to deduce, one can read it in the text or we see the colour of the landslide (red polygon). For better clarification we will write in the figure caption that the red colour represents activity state (active).

*315-320 'confidence degree': what is it?*

This is some kind of the evaluation of the achieved results. We evaluated that some landslides are active and we want to check if some evidence of this activity are present in the field. For more clarification we will try to rewrite this sentence.

*'measured displacements represent landslide dynamics': what does it mean? Landslide dynamic in the PS position? Or the entire landslide dynamic?*

We mean landslide activity state. This sentence will be corrected.

*'reality assessment': ??*

By reality assessment we want to prove/show evidence that activity state updated by using PS has the reflectance in the field. Since we don't have any other measurements (GNSS etc.), we could not evaluate our results direct. We can make it directly by the investigation/searching of the building and infrastructure damage. We interpreted this is a direct sign/indicator that specific landslide is active.

*'moderate damage': why this choice? This for sure can create a bias in the investigation*

We followed the recommendation and study of Mansour et al. (2011). They assess that landslide with velocity higher than 10mm will generate moderate damages to the buildings and infrastructure.

*'Activity states have been confirmed for 43 landslides': I guess the class, how did the expert confirmed the moderate class?*

We compare activity state (active vs. not active (dormant, stabilized)) by investigation of the damage. Thus, when we see the damage of a building in the specific time we can confirm that this landslide is active or not. We did not evaluate the level of the damage (moderate small etc.) or the level of intensity (slow moving or extremely moving) because for these we will need another data for instance (for intensity we will need GPS measurements or another levelling or total station that will give us quantitative information about the speed of the landslide). Thus, field validation is only used for the activity state verification.

*Landslide 'just-Tegoborze,' SOPO ID 23374*

*330 – 335 'Landslides tend to develop and increase activity over a large area.': the 23374 or in general?*

*This specific landslide called Just-Tegoborze with the id of 23374 in landslide database.*

*Landslide 'Zbyszyce' SOPO ID 73253*

*Landslide 'Liie-Jelna' SOPO ID 73194*

**360 – 365 'When reactivated, this landslide covered about 80% of the landslide area': what does it mean?**

*This means that some new movement were recorded for the 80% of the landslide body.*

*Landslide 'Wola Kurowska' SOPO ID 73254*

*Landslide 'Bartkowa-Posadowa' SOPO ID 72917*

*We left this SOPO id  because, the SOPO database is open and we added the link to this dataset. Therefore, when somebody is interested, can use the link type the landslide id and see this specific landslide in national landslide database.*

**4 Discussion**

**405-410 'landslide occurrence': Landslide occurrence or landslide monitoring? This sentence is a bad mix of different concepts.**

*We will improve this sentence.*

**'mostly connected': I suggest to relax the concept: 'can be largely connected...'**

*It will be corrected according to your recommendation*

**410 -415 'it was demonstrated that increasing the temporal sampling rate': I disagree, it was not formally demonstrated because the two series (12 and 6 days) don't overlap.**

*That's true that time span is not the same, but somehow based on PS density we can see the increase in PS density.*

**415 – 420 'the slope movement in a NE-SW direction represents only a small percentage of the real occurred displacements in LOS displacement rates...': This means that it is not only a matter of PS density… I think this point should have been taken into account more, both in the activity evaluation and in the discussion.**

*We will add this limitation into the text, probably (1) in the section of PS suitability analysis, (2) in section connected with results of the RI index and figure 4 when we present RI index for PS analysis. And (3) discussion section.*

**420-425 'homogenise landslide velocity': This is a (perhaps reasonable) assumption that needs to be verified. It can't come only from homogenization needs…**

**'real displacement rate can be much higher..': so how can a decision-maker trust and use this analysis?**

*The estimated velocity by using PS techniques cannot be lower for sure. But due to the ambiguous nature of the observation, if some "big" deformation appears (e.g. meters) then the coherence is lost and the real velocity cannot be estimated. This is the reason why this approach can only be used for slow to very slow-moving landslide. This is connected with underestimation so when PS shows as 100mm/year it can be a situation that there is a place within the landslide body with the speed of 200mm/year. So, we can have underestimated values but rather than overestimated values. Thus, it needs to be kept in mind that the real displacement can be higher than this estimated using PSI/*

*We will try to add this additional information when revising the manuscript.*

**430 – 435 'landslide activity states were evaluated by field': actually only a class was 'verified', so the evaluation is incomplete.**

*That true only a class active not active were evaluate. We will classify this issue.*

**I am not an expert in damages, so I can't say whether they were moderate or not.**

*As was previously mentioned, the expected range of damage was assessed based on Mounsour's thresholding, not based on our expertise. We will try to add extra information connected with this to be sure that reader knows that this was not evaluated by our "damage expertise".*

**'It can also support mentioned by Kroch..': I can't understand**

*We mean that Kroh  et al.,2017 also stated that despite the high landslide activity and density in this region there is still big pressure for urbanization and this should be minimized. We will try to clarify this when revising the manuscript.*

*Kroh, P. (2017). Analysis of land use in landslide affected areas along the Łososina Dolna Commune, the Outer Carpathians, Poland. Geomatics, Natural Hazards and Risk, 8(2), 863-875.*

**Conclusions**

---

## Author Comment (AC2) · 24 Jul 2020

*The authors are very grateful to the Editors and Associate Editors for the kind consideration and possible publication of our article in the Natural Hazards and Earth System Sciences. The authors would like to thank all reviewers for suggesting improvements for the manuscript. Point-wise reply/answer to each comment is provided below (comments are shown in* **green bold font**, *answers are shown as black font). All suggestions have been addressed, but still, the authors are open for further explanations and cooperation in term so manuscript improvement. Furthermore, the authors appreciate the editors and reviewers for the timely handling of the review process. After evaluation of this reviews we have the clear idea how to correct our manuscript to be publishable in NHESS. This mainly includes:*

- *Rewriting the introduction to highlight the novel aspect of the manuscript because in the present form, the novelty of the work is not clearly visible (rewriting according to the reviewer 2 suggestion)*
- *The literature review will be moved into material and methods sections (according to the reviewers 2 suggestion, shorten a little however will be kept because as the reviewer 3 stated this is a strong part of the introduction)*
- *According to the reviewer 2 suggestion, time series of the PS points will be presented together with the discussion about presented deformation mechanism*
- *According to the reviewer 1, some sentences will be rewritten to improve readably and some confusing aspect (thresholding, slope reprojection, PS technique limitation etc.) will be deeply discussed.*

*Authors*

**REVIEWER 2**

**POINT The authors have the data, but do not take a good advantage of them. In particular, they should show InSAR displacement time series to better demonstrate the state of activity of the studied landslides.**

*This will be added to the revised manuscript.*

**What is new in the study with respect to many (tens or even hundreds) other similar studies that used InSAR for landslide assessment? Why is your study important? The authors closely follow some papers published several years ago, largely outdated. There is nothing new in that, unless the authors would like to focus on the problems and provide a critical assessment of these kind of simplistic approaches. However, this requires considerable landslide expertize and good knowledge of local slope processes, which, I'm afraid, the author may not have. So, I think, to bring out some novelty they need to re-focus on the application limitations and on some potentially interesting aspects of the study (see below).**

*You are right that this methodology is applied by many authors. However, in the presented paper we would like to grab all various aspects in one comprehensive study.*

*Firstly, some authors use only one geometry of SAR images (ascending/descending) but here we used both. Additionally, in this study, we showed that using one SAR geometry is not appropriate approach due to the specific SAR geometry and terrain characteristic (slope/aspect). In some cases, ascending geometry is appropriate to investigate specific landslide and, in some cases descending images are required. Therefore, there is a need to investigate both geometries.*

*Moreover, in this study, we present some advantages of both Sentinel-1A/B data processing with 6-days revisiting time. This allows to increase the coherence and therefore increase the PS points density. This is a novel aspect of the paper, previously not presented- comparison between Sentinel-1A and Sentinel-1A/B data processing in terms of PS points density and therefore landslide activity state updating.*

*Furthermore, even though some paper presented already utilization of PSI for activity state updating, this is the first paper which shows landslide activity state updating in a region in Poland. This region is specific in comparison to study areas investigated in other papers (rural areas with extensive agriculture). This specificity will be emphasized in the revised version of the manuscript. Therefore, the results which are presented for another and characteristic study case may be interesting and beneficial for scientific community. Moreover, this is a first presentation of Sentinel-1 data application to estimate the landslide activity state in Polish Carpathians. The PS interferometry technique has been previously applied in monitoring Carpathian landslides by Perski et al., 2010, 2011. However, in different study area but due to the high temporal decorrelation, resulting from vegetation cover and short wavelength (X-band), it was only partially successful (Perski et al., 2009; Perski et al., 2011). This is mostly related to low PS density due to the TerraSAR-X data application. Therefore, exploitation of C-band (Sentinel-1) and L-band (ALOS PALSAR) data bring more advantages, especially in rural areas of Carpathians mountains and in our opinion this finding should be presented for scientific community. Moreover, inspired by reviewers, we will provide, in the revisited version, also critical discussion of the method used supported by accuracy analysis.*

*Perhaps we did not appropriately underline all novel aspects in introduction section of our manuscript. We will improve it while revising this manuscript.*

***InSAR results look OK. However, the assumptions regarding their use need to be better justified, limitations clearly acknowledged. The use of some rather arbitrary "thresholds" appears improper. Therefore, some of the interpretations and conclusions are questionable.***

*Thresholds which we have used in presented paper are widely applied in scientific community. We did not select these thresholds arbitrary. We followed many works with the same approach/thresholding for example: Bianchini et al. (2012); Bianchini et al., (2013) Cigna et al., (2013) Del Ventisette et al., (2014) or Barra et al., (2016) etc.*

***The manuscript is poorly organized, includes repetitions (for example, between part 1 and 2). Sloppy, even the sections and subsections of the manuscript are incorrectly numbered. Citations thrown in a haphazard fashion, often not representative or useful. Poor English.***

*Repetitions of some subsection has been made accidentally when we were adjusting our manuscript into the Copernicus NHESS template. This of course will be corrected in revised version of the manuscript. In our opinion, citations have been selected correctly however we will check one by one each citation to meet your suggestion. English has been checked by native spear before submission, however if reviewer stated that it is not appropriate, we will try to improved English in revised version of the manuscript and we will check it by native speaker again.*

***Detailed comments***

***Abstract – should improve. What are the relevant scientific questions you are trying to address??***

*Abstract as well as introduction will be rewritten and reorganized to underline novel aspects of this study.*

***1 Introduction This part includes some information about PSI technique that should be moved to part 2 Materials and Methods***

*This will be corrected to the revised manuscript. However, another reviewer stated that this is a strong part of the introduction, thus we will try to keep this comprehensive literature review in our manuscript.*

***At line 51-72 In recent decades, new methods for updating landslide inventory maps – too long, too many and outdated references.***

*This will be corrected of course. However, another reviewer stated that the advantage of the introduction is wide literature review. Thus, we will try to face both requirement and find the balance between these two reviews.*

*At line 65 Commonly used methodology in the abovementioned papers is the PSI matrix approach with diverse SAR sensors, where specific thresholding of the landslide velocity, acquired from specific PSI processing, are performed. – OK, but this belongs to Materials and Methods*

This can be corrected to the revised manuscript.

*Objectives – I suggest to focus more on the limitations and some potentially most interesting aspects of the study, that is: Evaluating the effect of SAR geometry delivered from ascending and descending orbits from ALOS PALSAR and Sentinel 1 and the sensitivity to measure deformation over the study area;*
*Evaluating the difference in landslide activity updated from three diverse data stacks, namely: L-band (ALOS), C-band with one satellite (Sentinel 1A, with a revisit interval of 12 days) and C-band with two satellites (Sentinel 1A and 1B with a revisit time of 6 days), respectively; â (New) Exploiting displacement time series for landslide activity/intensity assessment.*

Thank you for your suggestion. While evaluating the reviews, we think that we should totally rewrite introduction because some novel and interesting aspects were not properly highlighted, thus they were not clearly noticed. Thank you for noticing these aspects and for your recommendation. Based on this review, we concluded that we should focus on introduction improvement. Thus, for sure these aspects with SAR geometry of various sensors, time series of various landslides (active vs. not active or extremely slow vs very slow etc.) difference in landslide can be narrowed in introduction. This will be corrected to the revised manuscript.

*2 Materials and Methods This part mixes the description of the study area with the methods and includes unnecessary information like: This region is rural and, thanks to the breathtaking landscapes, the population has grown rapidly in recent decades. Moreover, it is also attractive to tourists.*

This will be removed from revised version of the manuscript.

*At Line 106: landslide distribution within the study area as well their predefined activity states. – what do you mean by predefined?*

Predefined means activity updated in 2010 by geologist during the field trip, when they were creating the official landslide database. Afterwards activity state has not been updated. An clarification to this issue will be added in revised manuscript.

*2.1.1 Geological and hydrological settings of the study area No hydrological information is included, while different names of formations and units are thrown in and these are nowhere shown.*

Using the term of "hydrological condition", we meant that Rożnów lake and Dunajec river undercut the slope which is the main reason of many landslide activation in this area. We will clarify this issue in the revised version of the manuscript

Different names of geological formation are presented in supplementary materials (it was placed in the appendix to increase readability of the manuscript)

*2.1.2 Landslide types and distribution (note that this section is repeated twice)*
*The landslide activity in the study area is mostly associated with hydro-geological conditions such as rock stratification and precipitation. These conditions created favorable conditions for landslide activation. – poorly written*

Repetition will be of course removed. This was included when adjusting style according to the NHESS requirements. English will be improved. We will include information about precipitation in the study area and we will try to correlate it with the activity state obtained using PSI based method. When revising our manuscript we will try to improve English and style.

*Most of the landslide scarps within the study area lead down to valley floors – not clear*

*This will be described more precise and clearer in revised version*

**2.1 Radar data and PSI processing (page 6)**

**At line 177 Therefore, exploitation of C-band (Sentinel-1) and L- band (ALOS PALSAR) data can bring more advantages, especially in rural areas (Lu et al., 2018). – seems like an incorrect reference as the work is about land deformation in Changzhou city and uses InSAR data sets from 2006 to 2012, that is before Sentinel-1**

*Thank you for this comment. Reference Lu et al., 2018 will be corrected in revised version.*

**2.2 PS post-processing phase (page 7)**

**After PSI processing, all results for the five diverse data sets have been post-processed in order to retrieve the most adequate displacement information – not clear what you mean by the most adequate, for what?**

*Adequate means that we performed SAR geometry analysis, reprojection into the deepest slope etc. This aspects allow us to remove some possibly not reliable PS points. This will be clarified in the revised version.*

**2.2.1 PS suitability analysis (page 7) – need to shorten as the information is already available in the literature**

*This will be shortened while revising the manuscript*

**At line 204 Therefore, conversion of LOS deformation into the most probable direction (direction of maximum slope), by assuming a pure translational movement mechanism, is commonly used (Bianchini et al., 2012). – hopefully not so commonly, as, for example you also have rotational or combined rock-debris slides in your study area. This need to be explained.**

*Yes, you are right however, many authors applied this simplified reprojection (Cigna et al., 2013). From our point of view, we can used LOS velocities, but then we cannot merge ascending and descending results together into the one database. Additionally, we can reproject it into the vertical and horizontal movement but real practice shows significant difficulties in mountainous and hilly areas where the chances of retrieving radar targets in both geometries (ascending and descending) are significantly lower due to geometrical visibility and distortions. Thus, using $v_{slope}$ seems a reasonable solution. Of course, this is as simplified transformation. We will provide an additional explanation when revising manuscript.*

**2.2.2 PS velocity projection along the steepest slope (page 8)**

**At line 232 Despite the great advantage of the motion represented in the slope direction, this projection has some limitations. First, when =90°, Vslope goes into infinity. Here we followed Herrera et al. (2013) and selected an absolute maximum value of - dangerous to talk about great advantage. Then, why should the approach by Herrera et al be good in your case?**

*It has the advantage that we can merge ascending and descending results together. Of course, in contrary this approach has disadvantage from the mathematical point of view.This approach was already applied by Bianchini et al. (2013) Kalia (2018). The C coefficient is the fraction of the 3D displacement that can be measured by PS targets and β the angle between the steepest slope and the LOS direction. Several limitations of this method must be taken into account: when β is almost 90°, C is close to 0 and $V_{SLOPE}$ tends to infinity. Following the work done by Herrera et al. 2013, an absolute maximum value of β = 72° corresponding to cos β = 0.3 is fixed and, as a result, $V_{SLOPE}$ cannot be higher than 3.33 times the $V_{LOS}$. This threshold corresponds to the condition number of 15 proposed by Cascini et al. 2010 as the number for the inversion matrix solving the algebraic system used for the projection process. In order to reduce any $V_{SLOPE}$ exaggeration, it is assumed C = −0.3 when −0.3 < C< 0 and C = 0.3 when 0 < C < 0.3. Whereas PS $V_{SLOPE}$ values turn positive ($V_{SLOPE}$ > 0) they are discarded. This is because a positive*

*V$_{SLOPE}$ would represent uphill movement. However, thank you for your comment, we will try to rewrite this sentence when revising our manuscript.*

**2.2.3 Velocity thresholding for activity state estimation - PSI based matrix approach (page 8)**
**At line 247 For the LOS velocity, distribution is almost normal (Gaussian), while for SLOPE is second negatively skewed as a result of the PS reduction (Bianchini et al., 2013). Therefore, for activity state estimation, we applied 5 mm/yr as the Vslope threshold. – not very clear. Why 5mm/yr? What is your measurement precision?**

*This threshold has been selected based on literature mentioned before. In order to directly evaluate the accuracy we need to have another data (e.g. levelling). Nevertheless, according to the internal accuracy estimation provided by SarScape software, PS velocity estimated by Sentinel-1 data is around 1mm/year and for ALOS is 3mm/year. We will add the information connected with accuracy estimation in the revised manuscript as follows:*

*Internal reliability assessment*

*Because lack of other data such as levelling etc., it was not possible to assess the accuracy of PSI approach. The effective estimation of displacement, atmospheric and other components of DInSAR, mainly depends on number of used acquisitions. Obviously, when a broad set of data stack is used, then more precise estimation of the results. Therefore, we used interior accuracy assessment based on Andrea Monti Guarnieri (2009)*

*Guarnieri, A.M. (2009) Accuratezza nella stima del rate di subsidenza. Politecnico di Milano*

*Assuming that we have N images, with temporal baselines Bt(i) with i = 1…N and normal baseline, Bn(i), we can define the following parameters:*

$$k_v = \frac{4\pi}{1000 * \lambda} \quad [\frac{rad}{mm}]$$

$$k_q = \frac{4\pi}{\lambda} * \frac{1}{Rsin\theta} \quad [\frac{rad}{m}]$$

*Where: λ - wavelength of the sensor, θ - incidence angle of scenes center, R - distance*

*Then, let define the matrix A*

$$A = \begin{bmatrix} 1 & k_qB_n(1) & k_v\frac{B_t(1)}{365.24} \\ 1 & k_qB_n(2) & k_v\frac{B_t(2)}{365.24} \\ ... & ... & ... \\ 1 & k_qB_n(N) & k_v\frac{B_t(N)}{365.24} \end{bmatrix}$$

*where the term 365.24 represents the average number of days in a year. Then we can evaluate the following matrix:*

$$C = (A^tA)^{-1}$$

*Note, if $(A^tA)$ is singular, or almost singular, it estimation of the velocity is impossible.*
*Standard deviation of the velocity estimation is:*

$\sigma_v \simeq \sqrt{c_{3,3} * \sigma_\phi} \ [mm/yr]$

*And $\sigma_\phi$ is calculated as:*

$$\sigma_\phi = \sqrt{\frac{\frac{4\pi^2}{\lambda^2}\sigma_r^2 - 2\log\gamma_S}{N_{PS}} - 2\log\gamma_P}$$

*Where*

$N_{PS}$ *– the average number of PS points per sq/km (in presented study, we assumed 200)*
$\gamma_P$ *-coherence of the target whose accuracy is to be assessed*
$\gamma_S$ *- the average coherence of the scene (typically $\gamma_S = 0.7$)*
$\sigma_r^2$ *- the standard deviation od two-way atmospheric disturbances (typically*
$\sigma_r^2 = (0.015)2)$
*For the final estimation of altitude precision of PS, we instead use the following formula:*

$$\sigma_v \simeq \sqrt{c_{2,2} * \sigma_\phi} \ [mm/yr]$$

*Presented formulas do not take into account global error such as phase unwrapping or orbital errors. For all these reasons, the formula can be assumed as the standard action of the best estimate, considering that in the real case this value it can get worse.*

***At line 251 Four diverse activity states have been determined (Fig. 5): (1) reactivated = active after being inactive, (2) active continuous = currently moving, (3) dormant = inactive, but possible to be reactivated and (4) stabilized = not active anymore. – You have InSAR data and should show displacement time series to clearly demonstrate the state of activity of your landslides. For example, are they continuously moving?? I doubt. Don't some of them stop in a dry season or winter?***

*You are right, in this region the driving force of landslide activity is precipitation. Since we performed our processing for time spans: 2014-2016 and year 2017, we estimated the activity for this time. Of course we will add time series deformation in revised manuscript and we will discussed this issue. Thank you for your suggestion.*

***2.2.4 Landslide intensity estimation (page 9)***
***3. Results (page 9) 3.1 Landslide activity state and intensity map generation – part of this section belongs to the previous one (2 Methodology).***

***At line 271 However, the activity state has been presented only for landslides where sufficient PS points have been found. At least four PS points within a landslide body were set up as the threshold. – why just four PS? I know that some authors you cite have indicated and used (also their followers) this arbitrary threshold, but is it scientifically sound? You may not justify its use.***

*For instance:*
*4 points within landslide body* → *Bianchini et al. (2013); Cigna et al. (2013)*
*3 points within landslide body* → *Cascini et al. (2013)*
*In the revised version we will implement the following improvement. Based on at least 4 points we will calculate the trimmed mean, removing two values; the smallest and the largest one. This will allow us to reduce outliers influence.*

***3.2. Possible hazard assessment (page 10) - part of this section belongs to the previous one (2 Methodology). However, why do you call it hazard assessment??? You are trying to assess possible damage. Moreover, please check the thresholds proposed by Mansour et al. (2011) really apply in your case and explain the limitations.***

*We will replace the word of hazard into the possible damage/destruction. This will be explained in revised version.*

***At line 304 Landslide with velocity below 10mm we classified as landslide with minor***

*expected damages. – questionable use of the threshold that can lead to dangerous interpretations. What if your landslide accelerates? And if t moves 9 mm/yr for 10 or more years?*

This actually very challenging topic with boundary values, which we meet very often in classification or everyday life. In case of classification the issue can be minimize by using natural jenks/breaks classification which allows us to increase difference between classes. Nevertheless, in this case the threshold has been assess by Monsour et al. (2011) based on correlation between destruction identify in field and movement.

*3.3 Field validation (page 11) – again, part of this section belongs to the previous one (2 Methodology). However, what do you validate in the field? You make some inferences based only on some simple observations. Moreover, you when to the filed 10 years after the ALOS data were acquired, and 1-few years after the acquisition of Sentinel data. Need to explain.*

Not really, field images were taken in various time. Unfortunately, we did not specify the dates of the photographs. We have tried to be in the field immediately after the time span of PS processing, see table below. We will include this information when revising manuscript.

| | Sensor used for activity evaluation in presented figure | Time span of analyzed data | Data of the field photos taken |
|---|---|---|---|
| Figure 11 | ALOS | 31/01/2008-27/12/2010 | 9/29/2010 |
| Figure 12 | Sentinel-1A+B | 2/01/2017-31/12/2017 | 03/07/2018 |
| Figure 13 | Sentinel-1A+B | 2/01/2017-31/12/2017 | 03/07/2018 |
| Figure 14 | ALOS | 31/01/2008-27/12/2010 | 25/03/2011 |
| Figure 15 | Sentinel-1A+B | 2/01/2017-31/12/2017 | 03/07/2018 |

*Figures Fig 1 (and others) – lack of coordinates; incomplete caption*

Caption and coordinates will be corrected when revising manuscript

*Fig 2 - > 60 geological units with no explanation, sending the reader to a supplementary material? You need to group them by similar lithology, with reference to the susceptibility to landsliding*

The geological unit will be merged together in revised version of the manuscript.

*Figs 4-6 Who needs to see these figures?*

*I thought that it will be better to imagine how this PSI based matrix approach works,*

*Fig 6 – Present or historical PSI data? Not clear what you mean.*

We will remove this figures, however we think that some aspects are better represented as an image rather than text. Similarly, other works present the PSI based matrix together with colors. This helps reader to connect specific PSI thresholding with specific activity. We will leave PSI velocity without word of "present or historical PSI data". Using this word, we thought that this velocity can be estimated by various PSI processing (actual sensors or past sensors). But of course we can remove it.

*C5 Fig 7 – overlapping colors, hard to see what is going on.*

It will be corrected however, the resolution and quality of the images has been degraded during pdf generation. We submitted also original images with high quality as a .zip package. Thus hopefully, images will be in better quality in final version of the manuscript.

*Fig 8 – Landslide intensity scale or simply velocity scale?*

*Based on Cigna et al. (2013) who deeply discussed PSI-based activity and intensity estimation approach, we used this terminology. Generally, intensity is already accepted term of movement speed or as you said velocity scale.*

*Fig 11 – wrong choice of color for the landslide area, as some PS have the same or very similar color*

*I will be corrected in the revised version of the manuscript*

---

## Author Comment (AC3) · 24 Jul 2020

*The authors are very grateful to the Editors and Associate Editors for the kind consideration and possible publication of our article in the Natural Hazards and Earth System Sciences. The authors would like to thank all reviewers for suggesting improvements for the manuscript. Point-wise reply/answer to each comment is provided below (comments are shown in* **green bold font**, *answers are shown as black font). All suggestions have been addressed, but still, the authors are open for further explanations and cooperation in term so manuscript improvement. Furthermore, the authors appreciate the editors and reviewers for the timely handling of the review process. After evaluation of the reviews we have the clear idea of the how to perform the corrections in order to improve our manuscript. This mainly includes:*

- *Rewriting the introduction in order to highlight the novel aspect of the manuscript because in the present form, the novelty of the work is not noticeable clearly visible (rewriting according to the reviewer 2 suggestion)*
- *The literature review will be moved into material and methods sections (according to the reviewers 2 suggestion, shorten a little however will be kept because as the reviewer 3 stated this is a strong part of the introduction)*
- *According to the reviewer 2 suggestion, time series of the PS points will be presented together with the discussion about presented deformation mechanism*
- *According to the reviewer 1, some sentences will be rewritten to improve readably and some confusing aspect (thresholding, slope reprojection, PS technique limitation etc.) will be deeply discussed.*

*Authors*

**REVIEWER 3**

***POINT 1 The main goal of this manuscript is to present the updated state of activity of landslides in Małopolskie municipality, a rural area with sparse urbanization in the Polish Flysch Carpathians, based on InSAR results. At a glance the manuscript seems to be a good case study, carried out according to state-of-the-art methodologies published and highly cited in the specialist literature. However, after a careful reading, it comes out that there is no element of novelty in both the InSAR processing & post-processing methodology and contribution to advance the field of landslide studies.***

*You are right that this methodology has been applied by many authors. However, in presented paper we would like to grab all various aspects in one comprehensive study.*
*Firstly, some authors use only one geometry of SAR images (ascending/descending) but here we used both. Additionally, in this study, we showed that using one SAR geometry is not appropriate approach due to the limitation of side looking SAR geometry and terrain characteristic (slope/aspect). In some cases, ascending geometry is appropriate to investigate specific landslide and, in some cases descending images are required rather than ascending ones. Therefore, there is a need to investigate both geometries.*
*Moreover, in this study, we presented some advantages of both Sentinel-1A/B data processing with 6-days revisiting time. This allows to increase a coherence and therefore increase the PS points density. This is a novel aspect of the paper, previously not presented- comparison between Sentinel-1A and Sentinel-1A/B data processing in terms of PS points density and therefore landslide activity state updating.*
*Furthermore, even though some paper presented already PSI utilization for activity state updating, this is the first paper which shows landslide activity state updating in a region in Poland. This region is specific in comparison to study areas investigated in other papers (rural areas with extensive*

*agriculture). This specificity will be emphasized in the revised version of the manuscript. Therefore, the results which are presented for another and characteristic study case may be interesting and beneficial for scientific community. Moreover, this is a first presentation of Sentinel-1 data application to estimate the landslide activity state in Polish Carpathians, (the PS interferometry technique has been previously applied in monitoring Carpathian landslides by Perski et al., 2010, 2011. However, in different study area but due to the high temporal decorrelation, resulting from vegetation cover and short wavelength (X-band), it was only partially successful (Perski et al., 2009; Perski et al., 2011). This is mostly related to low PS density due to the TerraSAR-X data application. Therefore, exploitation of C-band (Sentinel-1) and L- band (ALOS PALSAR) data bring more advantages, especially in rural areas of Carpathians mountains and in our opinion this finding should be presented for scientific community. Moreover, inspired by reviewers, we will provide, in the revisited version, also critical discussion of the method used supported by accuracy analysis.*

*Perhaps we did not appropriately underline all novel aspects in introduction section of our manuscript. We will improve it while revising this manuscript.*

**POINT 2 The area where this manuscript could bring in a novel contribution could be the improved knowledge about local landslides in Małopolskie and the risk that they pose to population, buildings and infrastructure.**

*As mentioned in previous point, we will try to rewrite introduction section in order to underline novel aspects of the manuscript, because due to not clear enough introduction, the novel aspect of the paper is not clearly noticeable. We will also try to extend discussion about landslide activity and risk in Małopolskie municipality according to your suggestion. Thank you.*

**POINT 3 However, despite the field validation, it is difficult to relate the conclusions achieved by the authors based on InSAR processing and analysis with the challenges that this area in the Polish Carpathians is experiencing. So in general I believe that this manuscript (and the research behind it) requires more work to be publishable. Further detailed comments are reported below.**

*According to the reviewer suggestion connected with issues related to landslide in Małopolskie municipality, in revised version of the manuscript, we can include additional analysis connected with building and infrastructure destructions investigated in field (there is an official report for the damage). We can compare this destruction map with estimated in this paper expected damage rate by Mansour et al. (2011),*

**POINT 4 Abstract It can be improved by removing redundancies (e.g. lines 12-13 vs 18), making explicit the cause for 7 landslides out of the total 50 that could not be confirmed, and highlighting the novel contribution with regard to either methodological approach or knowledge about local landslide issues and the risks that they pose (see also what the authors state at lines 84-87).**

*This issue will be corrected while revising the manuscript.*

**POINT 5 Introduction The strength of this section is for sure the wide and comprehensive literature review. Key and relevant papers are cited. However, although at line 72 the authors start listing the objectives of the manuscript, it is not clear how these objectives relate to the literature reviewed in the previous paragraphs. In which way is this manuscript novel (if it is) compared to the cited literature?**

*In our opinion, when compare our research with these listed in introduction section, we evaluated landslide activity state **in wider and comprehensive way.** Namely, many aspects have been jointly analyzed which in some papers are sometimes missing. To give more details about the difference between these works and our research we draw a couple of differences between our paper and peppers mentioned in introduction section.*

**DIFFERENCES WITH STUDIES MENTIONED IN LITERATURE**

*Bianchini et al. (2012) → (1) there ERS/Envisat data has been used and here we present deep investigation of various Sentinel-1 data; (2) Radar geometry and terrain orientation has not been evaluated and also velocity has not been reprojected into the slope direction (3) assessment of the possible hazard and damage has not been presented (4) field verification with damage evidences has not been shown.*

*Bianchini et al. (2013) → (1) they utilized ALOS data, besides ALOS data we applied also Sentinel-1 data for both satellites with different orbit geometries; (2) in contrary to that work, landslide intensity has been additionally evaluated and landslide damage map has been generated; (3) in contrary to that work, deep discussion about specific landslides, PS points and field investigation is presented.*

*Cigna et al. (2013) → (1) only descending images are utilized from ERS, Radarsat 1 and 2 in this paper while in our case ALOS ascending and Sentinel ascending and descending geometries are used (2) in contrary to that work, radar geometry and terrain orientation has not been evaluated (3) in contrary to our work, assessment of the possible hazard and damage has not been presented (4) in contrary to our work, field verification with damage evidences has not been shown, only google earth/google street map images*

*Del Ventisette et al. (2014) → (1) Envisat and ERS descending images have been used while in our case ALOS ascending and Sentinel ascending and descending geometries are used (2) in contrary to that work, landslide intensity has been additionally evaluated and landslide damage map produced; (3) in contrary to that work, we provide deep discussion about specific landslides, PS points and field investigations.*

*Barra et al. (2016).--> (1) 14 Sentinel-1 ascending images while in our case ALOS ascending and Sentinel ascending and descending geometries are used (2) in contrary to this work, different approach than PSI matrix has been utilized (3) in contrary to this work, no landslide intensity has been evaluated (4) in contrary to that work, field investigation has been performed and presented.*

*Kalia (2018) → (1) only 66 descending Sentinel-1 images while in our case ALOS ascending and Sentinel-1 ascending and descending geometries are used (2) in contrary to that work, assessment of the possible hazard and damages has not been presented (4) in contrary to that work, field verification with damage evidences have not been shown.*

**ADDITIONALLY, BESIDES THE DIFFERENCE BETWEEN MENTIONED PAPERS, THIS WORK PRESENTS:**

*-first wide and comprehensive exploitation of Sentinel-1 images when comparing to (Kalia, 2018; Monserrat et al., 2016; Barra et al., 2016)*

*-various aspects to improve reliability of PS points have been carried out (sensitivity index, reprojecting into slope direction)*

*-first landslide activity state evaluation in presented region in Poland. This region is unique and one of the most landslide affected areas in Poland.*

*-first Sentinel-1 data application in landslide activity state estimation in Polish Carpathians.*

*-first presentation of launching second Sentinel-1B satellite in terms of increase of PS points density*

*Maybe, we did not accurately describe and underline this novelty in the introduction section. Thus, we will improve the introduction to show these differences when revising our manuscript.*

**POINT 6 The feeling is that the authors primarily wanted to ensure that their methodology was aligned with the literature.**

*Through such extensive literature review connected with application of PSI based matrix approach we would like to provide current state of the knowledge rather than ensure our approach with these presented in literature. However, based on your previous comments, we will use this literature review to emphasize and to leverage our investigations. Thus, as mention previously, this issue will be improved in introduction when revising our manuscript and according to other reviewers' suggestion this literature review will be probably moved into the method section.*

**POINT 7 Figure 1: it is not clear whether the landslides mapped are those from the pre-existing inventory, i.e. prior to the update based on InSAR. The caption should be more explicative**

*These landslides are prior to the update based on InSAR. This is written as plain text in manuscript: "The study area covers the surrounding hills of Rożnów Lake (Fig. 1). This figure shows the landslide distribution within the study area as well their predefined activity states." Additionally, we will add this explanation in figure caption when revising our manuscript.*

**POINT 8 Methods The methodology is in general correct because it largely relies on well-established and accepted methodologies. However, it is difficult to see it framed within the specific geological, geomorphological and environmental features of the study area landscape. Therefore, although they may be correct, the rationale for the implementation of some of the assumptions is difficult to understand. At what extent the knowledge of the local landslides helped the authors to shape and adapt the methodology?**

*We use methodology being compilation of the methodologies presented by other authors dealing with landslide investigation by using PS interferometric processing. Based on Cigna et al., 2013, PSI-based methodology can be only applied for landslides with very slow dynamics, such as deep-seated gravitational slope deformations, creep, and rototranslational slides, flows, and complex landslides, as long as their velocities do not overcome the above-mentioned rates (Cigna et al., 2013). We investigated carefully, if landslide which we are dealing with are in this type of the movement. We investigated documentation from field investigation, which was created by well-experienced geologists in the study area. Based on this, we stated in the Materials and methods section that "Among the study area diverse landslide types exist including translational, rotational or combined rock-debris slides and typical debris slides". However, to improve our manuscript, we will add also information that this method can be used for slow moving landslide according to the previously mentioned literature and we will clarify that we investigated the type and speed of the movement of landslides within our study area.*

*You stated that the rationale for the methodology implementation is difficult to understand. However, this rationale is widely presented in literature and there are not many parameters which depend on*

"knowledge of the local landslide ". The fundamental "knowledge of the local landslide" depend on the velocity speed/intensity. This is strongly related with PSI interferometric limitation which is described in manuscript (not appropriate for fast movement). For this we investigated landslide documentation and characterization and based on this, we assessed that landslide within our study area are slow and extremely slow and therefore we can apply PSI based methodology. Besides this, we do not need any other geomorphological characteristics of the landslide etc. because PSI-based matrix approach depends of PSI-estimated velocity thresholding. This thresholding has already been used by abundant number of authors. Below, we present thresholds used by various authors and thresholds used in our study. Generally, when selecting the thresholding values, we followed the recommendations of widely cited papers, where also landslides are slow and extremely slow moving (please see below)

1) Landslide activity threshold
   $v_{LOS}$= -1.5mm Envisat, → Del Ventisette et al. (2014)
   $v_{LOS}$= -2mm, Envisat, → Bianchini et al. (2013)
   $v_{slope}$ = -5mm, Envisat, → Bianchini et al. (2013)
   $v_{slope}$ = -5mm, ERS, Radarsat 1&2, → Cigna et al. (2013)
   $v_{slope}$ = -10mm, Sentinel-1 → Kalia (2018)
   Since for $v_{slope}$ = -5mm is usually applied, we decided to use this threshold.
2) C value threshold of 0.3 (PS with C smaller that need to be removed)
   C=0.3 used by Kalia (2018)
   C=0.3 used by Bianchini et al. (2013)
3) Threshold to distinguish slow up to extremely slow mowing landslides

In general, in literature tree various threshold exists to distinguish slow moving and extremely slow-moving landslides.

Righini et al.2011 used vLOS=-10mm,
Bianchini et al. (2012) used vLOS=10mm
Cigna et al. (2013) used vslope =13mm.
Kalia (2018) used vslope =16mm.

As can be observed higher threshold are used in case of $v_{slope}$. (velocity reprojected into the steepest slope). In our study, we decided to used $v_{slope}$ =16mm similarly as Kalia (2018) but the most important reason is that this threshold is presented in well-know, old and widely respected literature (4218 citations) of Cruden and Varnes (1996). They stated that extremely slow landslide has velocity < 16 mm/yr and very slow landslides (16 mm/yr < velocity < 1.6 m/yr). Having considered abovementioned issues, we decided to applied threshold of $v_{slope}$ =16mm.

4) Number of points used to estimate velocity and
   4 points within landslide body → Bianchini et al. (2013); Cigna et al. (2013)
   3 points within landslide body → Cascini et al. (2013)

Based on the majority and to be more confident, we decided to used 4 points

5) way to estimated velocity average/clustering

average → Bianchini et al. (2012); Bianchini et al. (2013); Cigna et al. (2013)

clustering → Kalia (2018)

*Besides these parameters there is no other which need to be adjust to the geological or geomorphological conditions.*

*Summarizing, we will specify the issue connected with the "need of knowledge of local landslide characteristic" when revising our manuscript.*

**POINT 9 With regard to PSI, some key information are missing in the text (or I was not able to find myself), e.g. the location of the reference points selected during the processing.**

*Thank you for this comment, the location is really not included in text. We will, of course, add the reference point location in the figures 7 a,c,e where PS velocity is presented.*

**POINT 10 Figure 3: there is a typo in Feretti et al.; it should be Ferretti et al. Furthermore, in the text**

*These minor typos will be corrected during the manuscript revision.*

**POINT 11 I did not find the explanation of the rationale followed by the authors for processing first Sentinel-1A data only and then Sentinel-1A + Sentinel-1B data together. Would not it be better to process directly Sentinel-1A + Sentinel-1B data together? What is the advantage of this approach?**

*The rationale explanation to your question and our way of interferometric processing is that:*

(1) *Sentinel-1B data has been launched on April 24, 2016, thus Sentinel-1B data for the years 2014-2016 has not been available yet (only data from Sentinel-1A satellite are available for this period). When the second satellite Sentinel-1B has been launched (April 25th, 2016), it was available to use both satellites images in interferometric processing. Therefore, for year 2017 we used both satellites for PS processing.*

(2) *Additional objective of this study was to answer the question: What is the advantage of the launch Sentinel-1B satellite in terms of PS points density? Thanks to this strategy, we were able to answer this question in manuscript as "However, it was demonstrated that increasing the temporal sampling rate in view of launching the second Sentinel 1 satellite provides higher PS density and, ipso facto, more landslides could be investigated by means of PSI. We were able to investigate the landslide activity states of 130 and 205 landslides by using one Sentinel satellite and two satellites, respectively. It is significant progress in comparison to the first attempts with the application of PSI technique using TerraSAR-X and ERS-1 data in this study area (Perski et al., 2010, 2011) that failed due to disturbing vegetation cover" (lines 410-415)*
*Of course, we can marge these two data stacks into one interferometric processing, however we believe that these two processing strategies tangibly show that application of two satellite can increase PS density almost double it and therefore, we can get more information about deformation. This is direct recommendation to the scientific community dealing with PS processing, especially in rural areas with low coherence and small PS point density. In such area, we should take advantage of both satellites with a short revisiting time to get higher PS point density*

**POINT 12 More in general, what is the impact of each satellite dataset on the state of activity?**

*As previously described, these two separate PS processing (first by using from only Sentinel-1A image and second by using Sentinel-1A and 1B data), allow us to evaluate the effect of revisiting time on PS*

*coverage. When only Sentinel-1 data is used, revisiting time is 12 days while in second strategy is 6 days. This directly decrease temporal decorrelation and therefore increase coherence. Higher coherence means higher PS points density. Therefore, in conclusion section lines 227-250 we can find "In general, applying the PSI approach in rural areas is challenging. However, the results of this study prove that increasing the temporal sampling rate in view of launching the second Sentinel 1 satellite, provides higher PS density and, therefore, this technique can deliver useful information for landslide activity assessment. Sentinel 1A and Sentinel 1A/B were used for 130 and 205 landslides, respectively"*

*It can be observed in figure 9 where 20% of all landslide objects were updated by using Sentinel-1A while 38% where updated by using Sentinel-1A and 1B. This almost double increase is an important advantage.*

**POINT 13 Discussion I would have expected more linkage between the evidence gathered during the field validation and the InSAR data. When did the damages observed in the field happen? Is there a temporal association/correlation between the displacement showed by InSAR data and the damages?**

*Having observed Figures 11-15, where photos from field investigation were taken and PS points together with landslide extent are presented, we can notice a correlation between damages and PS points. Unfortunately, specific date of particular damage are not possible to be acquired because these need to make the interview with local inhabitants because such a damage cannot be dated based on remote sensing technique. In order to make more linkage between figures which represent PS points and photos take in the field, we present the table with time span if analyzed data and data when photos in the field have been taken. Unfortunately, this information has not been included in the manuscript, but if this will help to directly link field observation (photos) with PS results we will add this information in the revised manuscript of course.*

*Meanwhile, we have an access to official damage register for this region, however this is cumulative information, that is only roughly localized. Unfortunately, it cannot be used as the evidence for a particular landslide, but rather for the region.*

| | Sensor used for activity evaluation in presented figure | Time span of analyzed data | Data of the field photos taken |
|---|---|---|---|
| Figure 11 | ALOS | 31/01/2008-27/12/2010 | 9/29/2010 |
| Figure 12 | Sentinel-1A+B | 2/01/2017-31/12/2017 | 03/07/2018 |
| Figure 13 | Sentinel-1A+B | 2/01/2017-31/12/2017 | 03/07/2018 |
| Figure 14 | ALOS | 31/01/2008-27/12/2010 | 25/03/2011 |
| Figure 15 | Sentinel-1A+B | 2/01/2017-31/12/2017 | 03/07/2018 |

**POINT 14 Given the current statements at lines 432-436, it is not clear how the authors assessed the degree of damage to buildings and infrastructure.**

*Actually, it is written in manuscript in line 432 "All landslides presented as examples of field verification, based on Mansour's thresholding, are expected to generate moderate damage to buildings and infrastructure." Thus, we followed Mansour's thresholding which is explain in section 3.2 as:*

*"Based on a literature review, a downstream investigation was performed and additional thresholds were set up in order to assess possible hazards related to buildings and infrastructure located in landslide areas. For this purpose, we applied the method proposed by Mansour et al. (2011), i.e., the threshold of 10 to 100 mm/yr as a minimum landslide velocity which can cause moderate damage to infrastructure and buildings. Velocity rates higher than 100 mm/yr can cause major damage to infrastructure and buildings. Landslide with velocity below 10mm we classified as landslide with minor expected damages. This thresholding has been adopted as an additional criterion in order to support environmental planning and management strategies to areas which can be characterized by high landslide hazard and, consequently, should be addressed to potential damages protection. In Fig. 10, possible damages caused by mass movements in the study area are presented for three diverse PSI processing results." (lines 300-310)"*

**POINT 15 Several minor typos throughout the text need to be corrected, e.g. - line 16: "filed" to be corrected into "field" - line 181: "Feretti et al." to be corrected into "Ferretti et al."**

*These minor typos will be, of course, corrected during manuscript revision.*